# Mechanism of Arp2/3 complex branch disassembly by human Coro7

Nooshin Shatery Nejad [1,5], Malgorzata Boczkowska [2,5], Rouba Hilal [3,5], Fred E. Fregoso[2,4], Kyle R. Barrie [2,4], Grzegorz Rebowski [2], Andrew J. Saks [2], Alexis M. Gautreau [3], Enrique M. De La Cruz [1] & Roberto Dominguez [2,4] ✉

Arp2/3 complex nucleates branched actin networks that drive cell motility and intracellular trafficking. Coronins, a family of seven proteins in humans, inhibit Arp2/3 complex in vitro and reduce branch density in cells. Coro7, a distant member of this family, features two β-propeller domains (β1β2) and C-terminal Central-Acidic (CA) domains and remains poorly studied. Here, cryo-EM and biochemical data show that CA binds subunit Arp3 of free Arp2/3 complex with ~1 μM affinity, inhibiting polymerization like Arpin, while displacing Arp3's autoinhibitory C-terminal tail and promoting the active, short-pitch conformation, like WASP-family nucleation-promoting factors. Full-length Coro7, however, does not inhibit Arp2/3 complex polymerization but effectively induces debranching, whereas the isolated β1β2 or CA domains do not. In cells, Coro7 depletion disrupts ER-Golgi transport, which is rescued by full-length Coro7 but not by truncated variants. These results suggest that Coro7 functions as an Arp2/3 complex branch disassembly factor implicated in actin-dependent ER-Golgi trafficking.

Arp2/3 complex mediates the formation of branched actin filament networks essential for actin-based motility, including the movement of cells, intracellular organelles, and several pathogens[1]. It consists of two actin-related proteins (Arp2, Arp3) and five scaffolding subunits (ArpC1-5) and is regulated by several proteins. WASP-family nucleation-promoting factors (NPFs) bind to two sites on Arp2/3 complex, one on Arp2-ArpC1 and one on Arp3, triggering a conformational change toward the active, short-pitch state and delivering actin subunits at the barbed end of the Arps[2–6]. This prompts Arp2/3 complex to bind to the side of a pre-existing (mother) filament and initiate the formation of a branch (daughter) filament that grows from the barbed end of the Arps. Cortactin stabilizes the branch junction by binding subunit Arp3 of Arp2/3 complex and the daughter filament[7–9]. Another regulator, Arpin, also binds Arp3, but instead inhibits branch nucleation[10,11]. Finally, GMF binds subunit

Arp2 of Arp2/3 complex[12], preferentially in the ADP state[13], promoting debranching and network turnover[14–16].

Coronins represent another family of Arp2/3 complex regulators[17]. Humans express seven coronins, classified into three types: type 1 (coronins 1A, 1B, 1C, 6), type 2 (coronins 2A, 2B), and type 3 (coronin 7). Most studies on mammalian coronins have focused on types 1 and 2, which share a similar domain architecture, including an N-terminal β-propeller domain and a C-terminal trimerization coiled-coil domain. While these coronins, as well as *Saccharomyces cerevisiae* coronin (Crn1), inhibit Arp2/3 complex in vitro[18–20], their cellular activities are consistent with branch disassembly, as their depletion increases branch density, reduces network turnover, and impairs cell motility[19,21–23]. Mammalian coronin 7 (Coro7 hereafter) differs substantially from other coronins and remains poorly characterized. It features not one but two β-propeller domains, lacks the trimerization

[1]Department of Molecular Biophysics and Biochemistry, Yale University, New Haven, CT, USA. [2]Department of Physiology, Perelman School of Medicine, University of Pennsylvania, Philadelphia, PA, USA. [3]Laboratoire de Biologie Structurale de la Cellule (BIOC), CNRS, École Polytechnique, Institut Polytechnique de Paris, Palaiseau, France. [4]Biochemistry, Biophysics, and Chemical Biology Graduate Group, Perelman School of Medicine, University of Pennsylvania, Philadelphia, PA, USA. [5]These authors contributed equally: Nooshin Shatery Nejad, Malgorzata Boczkowska, Rouba Hilal. ✉e-mail: droberto@pennmedicine.upenn.edu

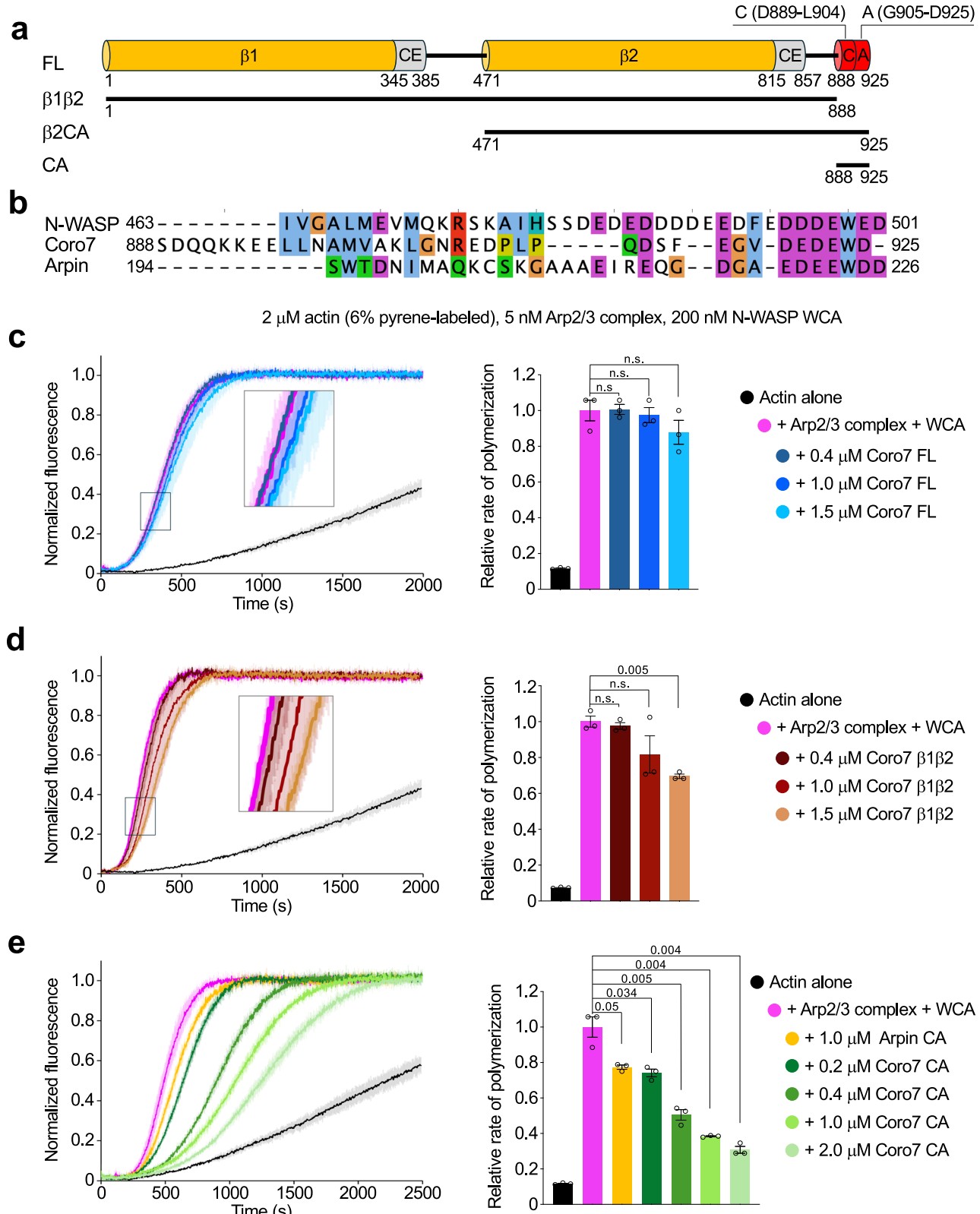

coiled-coil domain, and instead has C-terminal Central-Acidic (CA) domains, similar to those found in NPFs and Arpin (Fig. 1a, b). In cells, mammalian Coro7 associates with the Golgi network[24], in response to either Src-mediated phosphorylation[25] or E3 ligase-mediated polyubiquitination[26], and is thought to play a role in budding of Golgi-derived transport vesicles[27]. Although this activity has not been directly linked to Arp2/3 complex or debranching, a recent study on *C.*

*elegans* POD-1, a distant ortholog of human Coro7 (sharing 31% sequence identity), showed that it disassembles Arp2/3 complex branches in vitro, colocalizes with Arp2/3 complex at the cell leading edge, and plays a role in neuroblast migration[28].

Here, we characterize the structural-functional mechanism of Arp2/3 complex regulation by human Coro7. In a cryo-EM structure, the isolated CA region (Coro7 CA) binds subunit Arp3 of Arp2/3

**Fig. 1 | Effect of Coro7 constructs on Arp2/3 complex-mediated polymerization.**
**a** Domain diagram of Coro7 and description of constructs used in this study.
**b** Structure-based sequence alignment of the CA regions of mouse N-WASP (Uni-Prot: Q91YD9, PDB: 9DLX), human Arpin (UniProt: Q7Z6K5, PDB: 7JPN), and human Coro7 (UniProt: P57737, PDB: 9EAM, this work). Structurally equivalent amino acids were identified from a superimposition of subunit Arp3 across the three structures. For each sequence, the alignment begins with the first amino acid observed in the corresponding cryo-EM map. Coloring scheme: blue, hydrophobic; red, positively charged; magenta, negatively charged; green, polar; orange, glycine; yellow, pro-line; teal, histidine. **c**–**e** Left, time courses of actin polymerization by Arp2/3

complex (5 nM) activated by N-WASP WCA (200 nM), with and without varying concentrations of Coro7 constructs (FL, β1β2, or CA) or Arpin CA (1 μM). Actin alone (2 μM) is shown as a control. Data shown as the average curve from three independent experiments ± s.d. in lighter color. The inset shows a close-up view of curves that appear to overlap. Right, maximum polymerization rates calculated from $n = 3$ independent experiments, presented as mean ± SEM. For each data point, the $p$-values of pairwise comparisons with Arp2/3 complex + WCA (control) are listed (n.s., not significant, $p > 0.05$). Statistical significance was determined using an unpaired, two-tailed $t$-test with Welch's correction. Source data are provided in the Source Data file.

complex, occupying approximately the same site as the CA regions of NPFs and Arpin. Coro7 CA inhibits Arp2/3 complex nucleation like Arpin but promotes the short-pitch conformation like NPFs. In contrast, full-length Coro7 (Coro7 FL) does not inhibit Arp2/3 complex but exhibits strong debranching activity, directly visualized by microfluidics-TIRF microscopy, and promotes ER-Golgi trafficking of a glycosylphosphatidylinositol (GPI)-anchored reporter protein in cells. Both debranching in vitro and trafficking in cells require all Coro7 domains. The results suggest that Coro7 functions as an Arp2/3 complex branch disassembly factor in actin-dependent ER-Golgi trafficking.

## Results

### Coro7 CA inhibits Arp2/3 complex-mediated polymerization while Coro7 FL does not

Type 1 and 2 coronins inhibit Arp2/3 complex-dependent polymerization[18–20]. We asked whether Coro7 had a similar effect, for which several Coro7 constructs were tested (Fig. 1a). Constructs containing the β-propeller domains (FL, β1β2, and β2CA) were expressed in human Expi293F cells to ensure proper folding, whereas the C-terminal CA region, similar to those of NPFs and Arpin (Fig. 1b), was expressed in *E. coli*. Using the pyrene-actin polymerization assay[29], Coro7 FL and β1β2 did not significantly inhibit Arp2/3 complex polymerization activated by the WCA region of N-WASP (200 nM) and actin (2 μM) (Fig. 1c, d). In contrast, Coro7 CA strongly inhibited polymerization, and at a concentration of 1 μM, it was twice as inhibitory as Arpin CA (Fig. 1e), a known Arp2/3 complex inhibitor[10,11]. The inhibition of Arp2/3 complex by Coro7 CA, but not by Coro7 FL, suggests that the CA region is partially occluded within Coro7 FL. At higher concentrations, Coro7 FL and β1β2 showed a slight inhibitory trend, possibly due to competition with Arp2/3 complex for binding to the mother filament. This effect was more pronounced with Coro7 β1β2 than with Coro7 FL, suggesting that, like Coro7 CA, the F-actin-binding sites are partially occluded within Coro7 FL, possibly due to intramolecular interactions between CA, β1, and β2. Together, these observations seem to imply that in isolation, Coro7 FL exists in an autoinhibited state and may be activated at the branch junction (further addressed below).

### Coro7 CA binds to a single site on Arp2/3 complex and promotes the short-pitch conformation

Arpin binds Arp2/3 complex stoichiometrically and inhibits nucleation in two ways: (1) it competes with NPF binding to Arp3, and (2) it locks the C-terminal tail of Arp3 in the autoinhibited conformation, limiting the complex's ability to visit the active, short-pitch conformation[11]. We asked whether Coro7 CA shares these properties. Similar to Arpin CA[11], Coro7 CA bound both ATP- and ADP-Arp2/3 complex with approximately 1 μM affinity and 1:1 stoichiometry, as measured by isothermal titration calorimetry (Fig. 2a).

The effect of Coro7 CA on Arp2/3 complex conformation was analyzed using a cross-linking assay originally developed for budding yeast Arp2/3 complex[30]. Using our adapted version of this assay for recombinantly expressed human Arp2/3 complex[4], we monitored the increase over time of an Arp2-Arp3 cross-linking band, which can only

form when the Arps are aligned side-by-side in the short-pitch conformation (Fig. 2b, Supplementary Fig. 1). As previously observed[4], the Arp2/3 complex activator N-WASP WCA enhanced the intensity of the Arp2-Arp3 cross-linking band compared to control, consistent with an increase in the frequency with which Arp2/3 complex visits the short-pitch conformation. In contrast, the inhibitor Arpin CA limited this transition, also consistent with previous results[11]. Surprisingly, despite inhibiting Arp2/3 complex in the pyrene-actin assay (Fig. 1e), Coro7 CA promoted the short-pitch conformation like N-WASP WCA and contrary to Arpin CA. In the absence of further evidence, these two results seemed contradictory.

### Coro7 CA binds Arp3 at a site shared with NPFs, Arpin, and cortactin

To understand the source of the differences and similarities between Coro7 CA and both N-WASP WCA and Arpin CA, we determined the 3.0-Å resolution cryo-EM structure of human Coro7 CA bound to bovine brain Arp2/3 complex (Fig. 3a, Table 1, Supplementary Figs. 2, 3, Supplementary Video 1). The structure shows 38 amino acids of Coro7 CA (D889–D925) bound to a single site on Arp3. The central (or connecting) domain, corresponding to the N-terminal portion of Coro7 CA and referred to here as Coro7 C, consists of an amphipathic α-helix (D889–L904) that binds in the hydrophobic cleft at the barbed end of Arp3. Several hydrophobic side chains from this α-helix insert into the Arp3 cleft, including L896, L897, M900, V901, and L904. In Arp3, the cleft is lined by residues A150–L163 and F379–V393 on either side of Coro7 C, and by residues P412–F414 of the C-terminal tail of Arp3, which occupy the bottom of the cleft and interact with Coro7 C (Fig. 3g and Supplementary Video 1). The remaining C-terminal residues of Arp3 ($^{415}$GVMS$^{418}$) appear to be displaced by Coro7 C, causing them to become disordered.

Two other Arp2/3 complex regulators, WASP-family NPFs[4–6] and Arpin[11], have a central domain folded as an amphipathic α-helix, which, like that of Coro7, binds in the hydrophobic cleft of Arp3. However, the amino acid sequences and positions of the α-helices differ substantially in the three proteins (Figs. 1b, 3b–d). While the helices of Coro7 and NPFs overlap well and have similar lengths, that of Coro7 is shifted toward the back of the cleft, whereas that of NPFs is shifted toward the front, such that they only overlap for two helical turns (Fig. 3c). Yet, they both displace the autoinhibitory C-terminal tail of Arp3 (Fig. 3f–h), which is thought to be one of the ways by which NPFs promote the short-pitch conformation[4,6,31,32]. In contrast, Arpin has a shorter helix, which binds toward the front of the cleft (Fig. 3d), such that it does not compete with the C-terminal tail of Arp3. Instead, Arpin stabilizes the inhibited conformation of the Arp3 tail through hydrophobic contacts between W196 within Arpin's α-helix and residues V413 and F414 of Arp3's C-terminal tail (Fig. 3f, i)[11].

The acidic domain, corresponding to Coro7 C-terminal residues G905–D925 and referred to here as Coro7 A, binds along the side of subdomains 3 and 4 of Arp3 (Fig. 3a). Several basic amino acids line the path of Coro7 A on Arp3, including K244, K251, R275, R329, R333, R334, R337, and R341. Within Coro7 A, the motif $^{920}$DEDEWD$^{925}$, which harbors the conserved tryptophan W924, binds in a pocket formed by Arp3 residues R329–R341 and Y233–K244 (Supplementary Video 1).

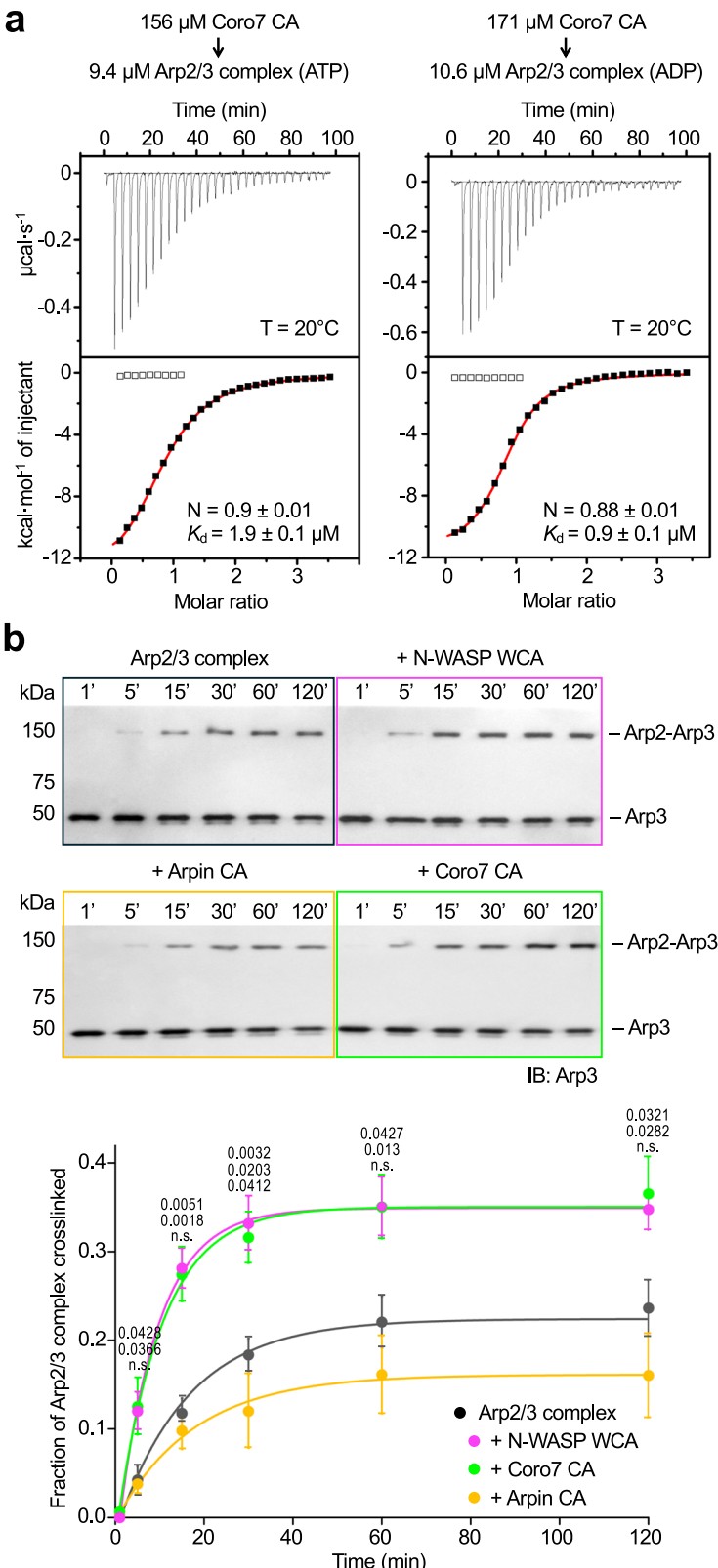

This binding site is shared by several Arp2/3 complex regulators that have a related acidic domain and a conserved tryptophan, including WASP-family NPFs[4–6], Arpin[11], and cortactin[8,9] (Fig. 3e). The acidic domain is found at the C-terminus of these proteins, except in cortactin, where it is found at the N-terminus. Probably as a result, cortactin binds to the pocket in Arp3 with inverted polypeptide polarity compared to the other proteins in this group[8,9]. Since the binding sites of Coro7, NPFs, Arpin, and cortactin on Arp3 partially overlap, these four regulators are likely to compete for binding to Arp2/3 complex.

In summary, the structure suggests that Coro7 CA promotes the short-pitch conformation by displacing Arp3's C-terminal tail like NPFs while competitively inhibiting NPF binding to Arp3 like Arpin[11], a necessary step in Arp2/3 complex activation[4].

**Fig. 2 | Coro7 CA binds to a single site on Arp2/3 complex and promotes the short-pitch conformation. a** ITC titration of Coro7 CA (in the syringe) into ATP- or ADP-bound Arp2/3 complex (in the cell). Experimental conditions, including temperature (T) and protein concentrations, and fitting parameters, including stoichiometry (N) and dissociation constant ($K_d$), are listed. **b** Western blot analysis (top) and densitometric quantification (bottom) of the crosslinked fraction of Arp2/3 complex under various conditions (Arp2/3 complex ± WCA: $n = 3$, Arp2/3 complex + Arpin CA: $n = 4$, Arp2/3 complex + Coro7 CA: $n = 6$, see also Supplementary Fig. 1). An anti-Arp3 antibody was used to identify the crosslinked Arp2-

Arp3 band, which only forms in the short-pitch conformation. Experiments with N-WASP WCA, Coro7 CA, or Arpin CA are represented by black, magenta, green, and orange gel outlines and graph traces, respectively. Data are presented as mean ± SEM from $n$ independent experiments and were fitted to a first-order exponential equation. Statistical significance was determined using an unpaired, two-tailed $t$-test with Welch's correction. For each data point, $p$-values for pairwise comparisons with Arp2/3 complex alone (control) are shown from top to bottom for experiments with N-WASP WCA, Coro7 CA, and Arpin CA (n.s., not significant, $p > 0.05$). Source data and fitting parameters are provided in the Source Data file.

## Coro7 FL dissociates Arp2/3 complex branches

The β-propeller domains of various coronin-family members, including *Drosophila* POD-1 and *Dictyostelium* Coro7, have been shown to bind F-actin[33–38], a property found here to be shared by human Coro7 (Supplementary Fig. 4a). Interestingly, however, in cosedimentation assays with F-actin, Coro7 FL, β1β2, β1, and β2 all bound with similar apparent affinities, ranging from 5 to 11 μM. While affinities derived from cosedimentation and gel quantification have inherent limitations, the data suggest that β1 and β2 do not contribute additively to F-actin binding by Coro7 FL, possibly because their actin-binding surfaces are at least partially occluded within the full-length protein. This is also consistent with the observation that Coro7 CA, but not Coro7 FL, inhibits polymerization induced by Arp2/3 complex (Fig. 1c, e). Furthermore, contrary to Coro1B, which displays higher affinity for F-actin in the ADP-Pi than in the ADP state[34], Coro7 FL had ~3.5-fold higher affinity for F-actin in the ADP state than in the ADP-BeF$_3^-$ state (Supplementary Fig. 4b), consistent with it targeting older branches (after Pi release).

Above, we demonstrated that Coro7 CA binds Arp2/3 complex and promotes the short-pitch conformation (Fig. 2), which is the conformation observed at the branch junction[39]. The combined ability of Coro7 to bind F-actin and Arp2/3 complex suggests that it may specifically target the branch junction, either for debranching, like other coronins[19,21–23,28], or stabilization, like cortactin[9]. To test these possibilities, we analyzed the effect of Coro7 constructs on the dissociation of filament branches under constant force using a microfluidics apparatus and direct visualization by TIRF microscopy[15]. Control experiments conducted at a flow rate of 25 μL min$^{-1}$, corresponding to an estimated force on branches of 0.05 pN, showed that most branches remained after 13 min in the absence of Coro7 FL, whereas a substantial fraction dissociated in the presence of 70 nM Coro7 FL (Supplementary Fig. 5a). To accelerate branch disassembly, most experiments were thus conducted at the higher flow rate of 200 μL min$^{-1}$, corresponding to an estimated force of 0.45 pN, and either in the absence or the presence of Coro7 constructs (Fig. 4a).

In the absence of Coro7 FL, the debranching time courses followed single exponentials, with an observed rate constant ($k_{obs}$) of ~0.003 s$^{-1}$ (Fig. 4a–c, Supplementary Video 2). In contrast, in the presence of various concentrations of Coro7 FL, the debranching time courses followed double exponentials (Fig. 4b, Supplementary Video 3), indicating that debranching in this case follows a different pathway, described by the following equations:

$$BM \overset{k_{debranch}}{\rightleftharpoons} B + M$$
$$\updownarrow K_{Coro7}$$
$$CBM \overset{k_{Coro7,\ debranch}}{\rightleftharpoons} (C)B + (C)M$$

where $B$, $M$, and $C$ stand for branch filament, mother filament, and Coro7 FL, respectively, $K_{Coro7}$ is the binding affinity of Coro7 FL for the branch junction, and the parentheses around $C$ indicate that Coro7 FL may remain bound to either $B$ or $M$ after debranching. The $k_{obs}$ of the slow phase was the same as in the absence of Coro7 FL (~0.003 s$^{-1}$),

which we interpret as branch dissociation through the top pathway ($k_{debranch}$), whereas that of the fast phase was ~10 times greater (~0.03 s$^{-1}$), consistent with accelerated debranching by Coro7 FL through the bottom pathway ($k_{Coro7,\ debranch}$). The fast phase was more force-sensitive than the slow phase (Supplementary Fig. 5c), indicating that branches rupture over a narrower force range in the presence of Coro7 FL[40]. The observed rate constants of the fast and slow phases were independent of the concentration of Coro7 FL (Fig. 4c), but the relative amplitudes of the fast-debranching phase depended hyperbolically on the concentration, resulting in an apparent affinity of Coro7 for the branch junction ($K_{Coro7}$) of ~30 nM from the best fit of the data to a rectangular hyperbola (Fig. 4d).

Debranching was also accelerated by Coro7 β2CA, albeit to a lesser extent (Fig. 4e, Supplementary Video 4), but not by Coro7 β1β2 (Fig. 4f, Supplementary Video 5) or Coro7 CA (Fig. 4g, Supplementary Video 6), which lack the Arp2/3 complex- and F-actin-binding regions, respectively. These results suggest that human Coro7, including all its domains, functions as a debranching factor rather than an Arp2/3 complex inhibitor, with the latter activity observed only with the isolated CA region (Fig. 1e).

## Coro7 depletion disrupts ER-Golgi transport

Human Coro7 has been localized to the Golgi network and implicated in cargo sorting and export[24,25,27,41]. Building on these observations, we imaged here the secretory pathway in an RPE-1 stable cell line expressing the GPI-anchored reporter GFP-GPI, combined with electroporation of Coro7 variants into a CRISPR/Cas9-generated Coro7 knockout (KO) background (Fig. 5, "Methods"). The RUSH system was used to enable biotin-induced, synchronous release of ER-accumulated GFP-GPI[42]. Upon release ($t = 0$), the fluorescent reporter first accumulates in the perinuclear region of the Golgi, before exiting and fusing with the plasma membrane (Fig. 5d, Supplementary Video 7). We quantified the condensation time of the fluorescence signal at the Golgi, which, unlike the signal at the plasma membrane, is synchronized.

In Coro7 KO cells, the GFP-GPI signal condensed at the Golgi with a significant delay compared to parental cells (15.1 vs. 23.8 min) (Fig. 5d, e and Supplementary Video 7). Attempts to rescue the KO phenotype by transfecting cells with a plasmid expressing Coro7 FL failed due to excessive and uncontrollable overexpression. Thus, we used electroporation of Flag-iRFP-tagged Coro7 protein variants (FL, β1β2, and β2CA) purified from 293T cells (Fig. 5a), ensuring the amount of protein introduced approximately matched the endogenous level of Coro7 (Fig. 5b, c, Supplementary Fig. 6). Electroporated FL Coro7 rescued the delayed trafficking of GFP-GPI in Coro7 KO cells (condensation time: $t = 16.4 ± 2.9$ min), while constructs Coro7 β1β2 and Coro7 β2CA failed to rescue this phenotype (Fig. 5d–f). These results suggest that, similar to debranching in vitro (Fig. 4), all Coro7 domains are required for efficient ER-Golgi trafficking in cells, including the filament-binding β-propeller domains and the Arp2/3 complex-binding CA region.

These observations, however, did not directly link the Coro7 KO phenotype to defective debranching during ER-to-Golgi transport. To address this, we co-imaged cortactin, a marker of branched actin

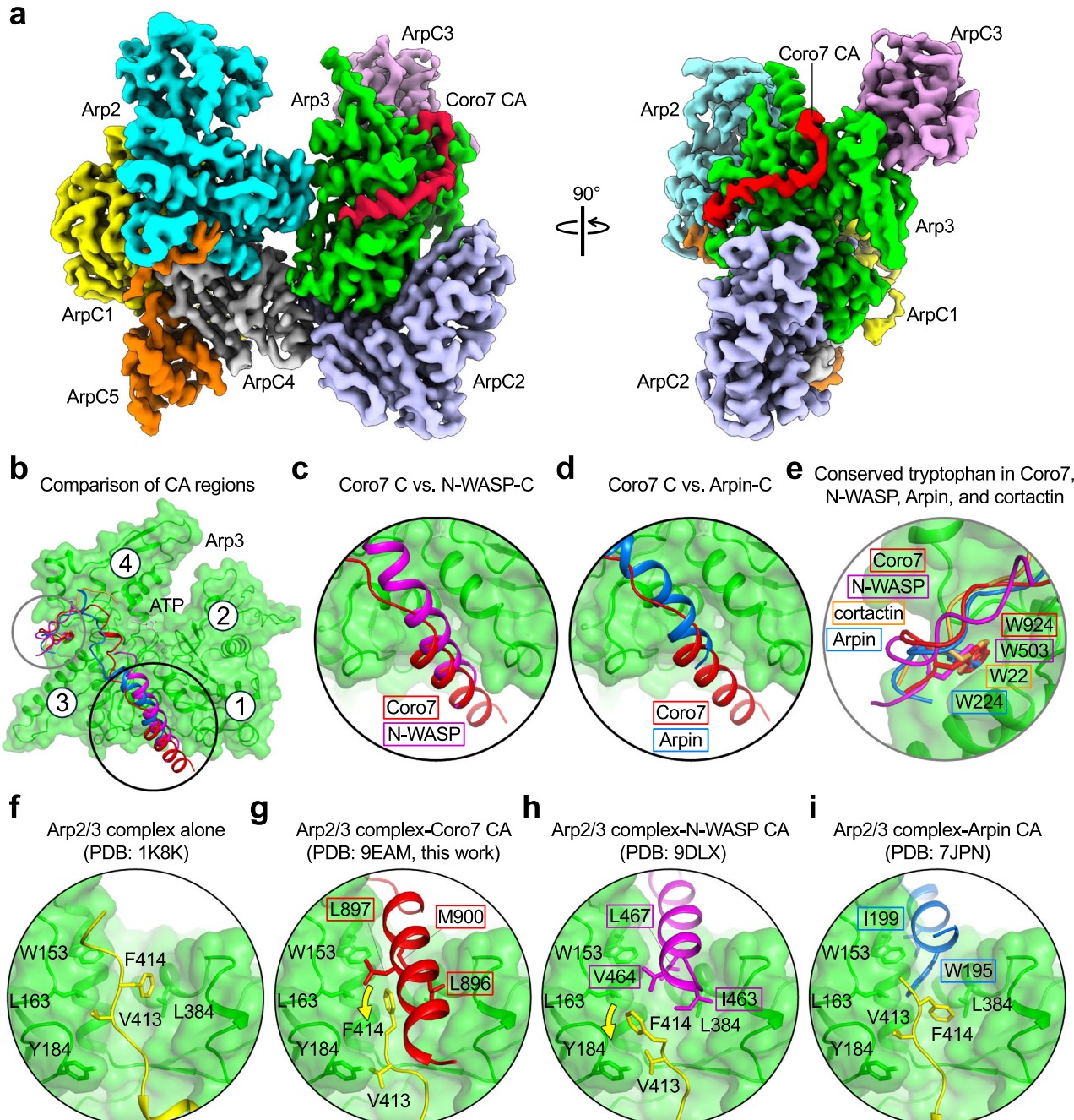

**Fig. 3 | Coro7 CA binds Arp3 at a site shared with NPFs, Arpin, and cortactin.**
**a** Two perpendicular views of the cryo-EM map of Coro7 CA bound to Arp2/3 complex, colored by protein subunits. The entire Coro7 CA region is visualized in the map (red). **b** Comparisons of the structures of Coro7 CA (red) with the equivalent region of N-WASP (magenta), Arpin (blue), and cortactin (orange). **c**–**e** Close-ups showing comparisons of the amphipathic α-helix of Coro7 C, which binds in the hydrophobic cleft of Arp3, with those of N-WASP (**c**) and Arpin (**d**), as well as the region around the conserved tryptophan W924 of Coro7 A with those of N-WASP, Arpin, and cortactin (**e**). **f**–**i** Close-ups showing the hydrophobic cleft

(green) and autoinhibitory C-terminal tail (yellow) of Arp3 in the structures of (from left to right) inactive Arp2/3 complex (PDB: 1K8K) and Arp2/3 complex with bound Coro7 CA (PDB: 9EAM, this work), N-WASP WCA (PDB: 9DLX), and Arpin CA (PDB: 7JPN). Notice that the α-helices in the C regions of both N-WASP and Coro7 displace Arp3's C-terminal tail from its position in the inactive complex (indicated by a curved yellow arrow). In contrast, Arpin residue W195 interacts with hydrophobic residues of Arp3's C-terminal tail (V413 and F414), stabilizing its autoinhibited conformation.

networks, and pVenus-GBF1, associated with the endoplasmic reticulum–Golgi intermediate compartment (ERGIC)[43], in WT and Coro7 KO MCF10A cells. Cortactin immunostaining co-localized with pVenus-GBF1 in the perinuclear region of MCF10A cells and was markedly increased upon Coro7 depletion, consistent with reduced debranching activity (Supplementary Fig. 7). Together with our observation that GFP-GPI trafficking through the ERGIC is substantially

delayed in Coro7 KO RPE-1 cells, these findings support the notion that ER-to-Golgi transport is impaired in the absence of Coro7 due to defective debranching.

## Discussion
Cellular studies have implicated all coronins studied thus far in debranching and recycling Arp2/3 complex networks[19,21–23,28]. Our results

**Table 1 | Cryo-EM data collection and statistics**

| Cryo-EM data collection | |
|---|---|
| Microscope | Titan Krios G3 (NCEF) |
| Camera | K3 Summit |
| Acquisition | EPU software |
| Magnification | ×105,000 |
| Voltage, keV | 300 |
| Defocus range, µm | −0.5 to −2.5 |
| Pixel size, Å | 0.428 (super-resolution mode) |
| Electron exposure, e⁻/Å² | 50 |
| Movies | 7295 |
| Initial no. particles | 241,121 |
| **Map** | |
| Symmetry | C1 |
| Final no. particles | 116,851 |
| Map resolution, FSC threshold 0.143 (0.5), Å | 3.0 (3.3) |
| Resolution range, Å | 2.7–43.0 |
| **Refinement and validation** | |
| Initial models used (PDBs) | 8TAH |
| Model resolution, FSC threshold 0.5, Å | 3.2 |
| Sharpening B factor, Å² | −108.3 |
| **Model composition** | |
| No. non-hydrogen atoms | 15,778 |
| No. residues | 1967 |
| No. ligands | ATP:2, MG:2 |
| **Correlation model vs. data** | |
| CC (mask, volume) | 0.90, 0.89 |
| CC (ligands) | 0.91 |
| **r.m.s deviations** | |
| Bond lengths, Å (no. > 4σ) | 0.005 (0) |
| Bond angles, ° (no. > 4σ) | 0.855 (18) |
| **Validation** | |
| MolProbity score | 1.02 |
| Clashscore | 1.24 |
| Rotamer outliers, % | 0 |
| **Ramachandran plot** | |
| Favored, % | 96.97 |
| Allowed, % | 2.92 |
| Disallowed, % | 0.1 |
| **ADP (B-factors)** | |
| Protein, Å² (min/max/mean) | 72 / 298 / 144 |
| Ligand, Å² (min/max/mean) | 65 / 161 / 133 |
| **Accession codes** | |
| EMDB | EMD-47836, EMD-47408, EMD-47746, EMD-47769, EMD-47770, EMD-47772, EMD-47795, EMD-47797 |
| PDB | 9EAM |

suggest that mammalian Coro7 shares this role, despite its domain organization differing substantially from other coronin-family members. Our analysis also indicates that all Coro7 domains must be considered together to understand its cellular function. While dissecting the mechanisms of Arp2/3 complex inhibition by Coro7 CA (Figs. 1–3) and F-actin binding by the β-propeller domains (Supplementary Fig. 4) was informative, these are unlikely to represent cellular activities of Coro7 FL, which likely does not target either Arp2/3 complex or F-actin

in cells. Indeed, binding to F-actin or the isolated Arp2/3 complex would result in nonproductive interactions. On the other hand, neither Coro7 CA nor β1β2 affected branch stability in isolation. Both microfluidics-TIRF (Fig. 4) and cellular (Fig. 5 and Supplementary Fig. 7) experiments showed that all Coro7 domains must work synergistically to target branch junctions and promote their disassembly during ER-to-Golgi trafficking. Debranching was also previously observed with *C. elegans* POD-1[28], indicating that this activity is conserved across Coro7-family members. Curiously, however, these authors found that the CA region was dispensable for debranching. It remains to be demonstrated whether this discrepancy reflects differences between POD-1 and human Coro7, which are only distantly related.

Coro7 FL did not inhibit Arp2/3 complex polymerization (Fig. 1c) and the β-propeller domains did not contribute additively to F-actin binding (Supplementary Fig. 4a), suggesting that the Arp2/3 complex- and F-actin-binding CA and β1β2 regions are partially occluded within the full-length protein. At branch junctions, Coro7 may undergo a conformational change, driven by the combined affinities of its various domains for short-pitch Arp2/3 complex, F-actin, and possibly the interface between them, enabling it to specifically target the junction. We also found that, contrary to Coro1B[34], Coro7 FL has ~3.5-fold lower affinity for F-actin in the ADP-BeF$_3^-$ state than in the ADP state (Supplementary Fig. 4b). This is consistent with the expectation that debranching targets older branches, after Pi release.

Human Coro7 has been implicated in secretory trafficking through the Golgi[24,25,27,41], which aligns with our findings (Fig. 5 and Supplementary Fig. 7). We further showed that all Coro7 domains, required for efficient Arp2/3 complex branch dissociation in vitro, are also critical for promoting efficient trafficking of secretory cargo from the ER to the Golgi in cells. Arp2/3 complex, activated by the NPF WHAMM, has been previously linked to the generation of branched actin networks along this secretory route[44]. Together, these observations suggest that Coro7 plays a role in branched network disassembly along the ER-Golgi secretory pathway, analogous to how coronin 1B disassembles branched networks in lamellipodia[21].

Collectively, the results obtained here allow us to propose a debranching model for Coro7 (Fig. 6, Supplementary Fig. 8, Supplementary Video 8). We began by fitting the structure of Coro7 CA onto the Arp3 subunit of Arp2/3 complex at the branch, using the structure of the cortactin-bound branch junction (PDB: 8P94) as a reference[9]. This suggested that the most likely binding site of β2 is at the interface between the first two actin subunits along the long-pitch helix at the barbed end of Arp3. Three key observations support this hypothesis. First, the linker between β2 and CA is short (23 amino acids, P864-K887) allowing only for two potential binding sites for β2: the proposed site and a site on the mother filament, which was excluded due to a small clash with Arp2/3 complex (Supplementary Fig. 8e). Second, the proposed position of β2 aligns well with the 8.6-Å resolution cryo-EM map of the β-propeller domain of yeast coronin 1 (Crn1) bound to F-actin[38] (Supplementary Fig. 8a–c). Third, AlphaFold3[45] positions both Coro7 β1 and β2 at the same interface between actin subunits when either domain is modeled with seven actin subunits (Supplementary Fig. 8a). Notably, the AlphaFold3 models of β1 and β2 only need to be shifted laterally by ~3 Å to fit the cryo-EM map of Crn1 (Supplementary Fig. 8c). Given this agreement, we adopted the β-propeller positions suggested by AlphaFold3, predicted based on the correct sequence of human Coro7 (Supplementary Fig. 8).

Once β2 and CA are positioned, the location of β1 becomes less deterministic, as the linker between β1 and β2 is long (71 amino acids, P401-S471), allowing for multiple potential binding sites (Supplementary Fig. 8f). As with β2, we excluded the only available binding site on the mother filament due to a small clash with Arp2/3 complex. However, on the daughter filament, β1 could potentially bind at three or more different locations on either side of the filament. One of these

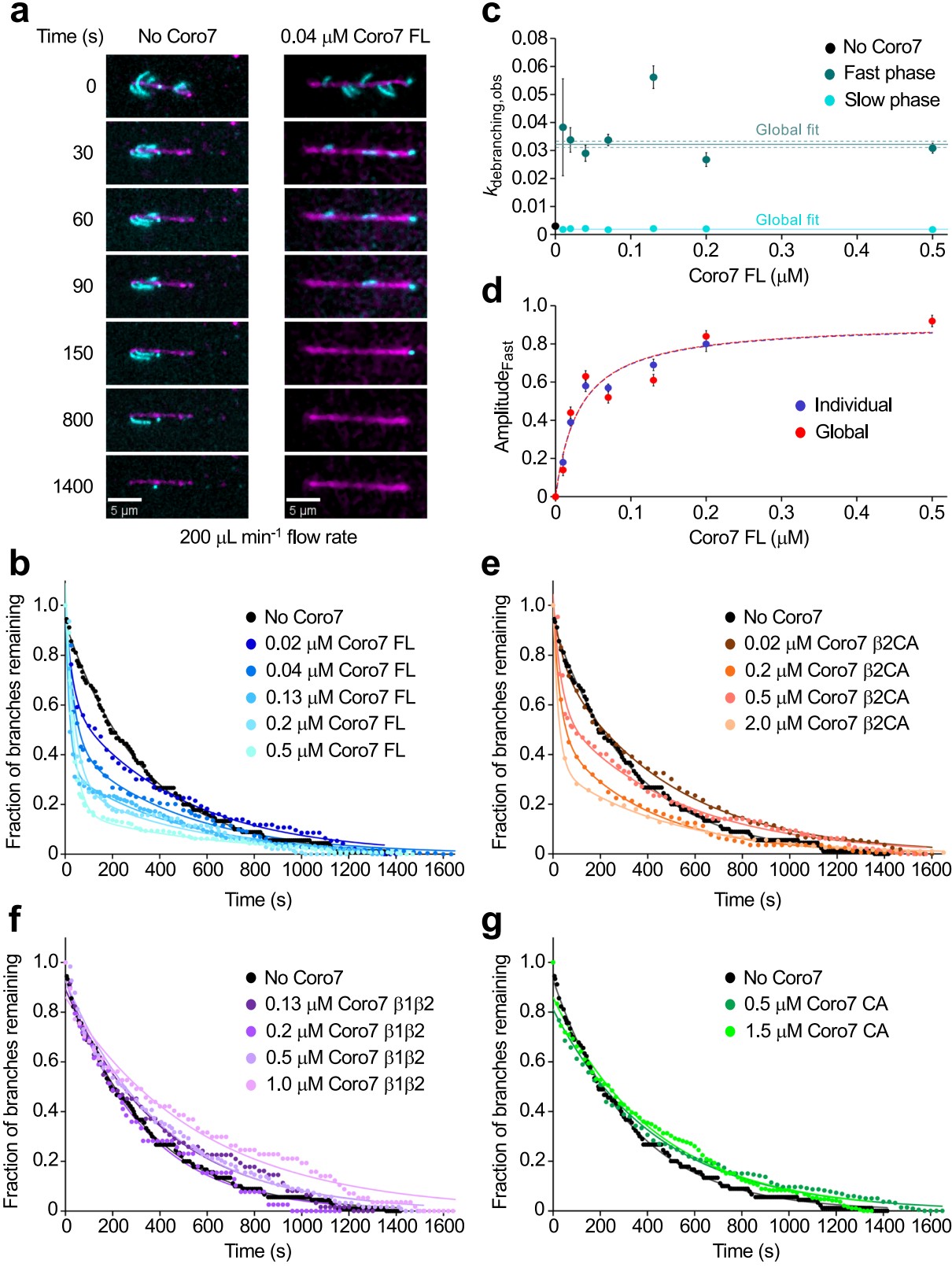

sites is at the interface between Arp2 and the actin subunit bound at its barbed end, which we favor, as it would imply β1 specifically targets the branch junction (Supplementary Fig. 8f). This may also explain why Coro7 FL has higher debranching activity than β2CA (Fig. 4).

We hypothesize that Coro7 functions as a branch-specific severing protein, forming a pincer at the interface between Arp2/3 complex and the daughter filament, with β1 and β2CA acting as the "claws" of this pincer on each side of the filament. The binding of the α-helix of Coro7 C in the hydrophobic cleft of Arp3 is incompatible with both the binding of Arp3's autoinhibitory C-terminal tail (Fig. 3g) and the binding of the D-loop of the actin subunit at the barbed end of Arp3 (Supplementary Fig. 8e and Supplementary Video 8). This could serve as the first site of attack for Coro7, destabilizing the interface between Arp2/3 complex and the daughter filament. On the opposite side of the

**Fig. 4 | Branch dissociation by Coro7. a** Time-lapse TIRF microscopy images of debranching events in the absence (left) or presence (right) of Coro7 FL. The flow rate is 200 μL min⁻¹, corresponding to an average force of -0.45 pN on an average -2 μm-long branch[15]. **b** Time courses of branch dissociation in the absence or presence of Coro7 FL. Solid lines through the data represent best fits to single (without Coro7 FL, black) or double (with Coro7 FL, blue shades) exponentials. **c** Debranching rate constants as a function of Coro7 FL concentration. The individual data points represent the observed debranching rate constants obtained from the best fits of the time courses shown in (**b**) to double exponentials. The teal and cyan horizontal lines represent the best global fits of the debranching time courses shown in (**b**) to a double exponential with a single rate constant for each phase (fast and slow). **d** Dependence of the fast-debranching phase amplitudes on Coro7 FL concentration. The blue circles represent the amplitudes obtained from individual fitting of

the time courses shown in (**b**) to double exponentials, and the red circles represent the relative amplitudes obtained from global fitting. The solid line through the data represents the best fit to a rectangular hyperbola, yielding an apparent Coro7 FL binding affinity of 31 ± 1 nM for both individually and globally fitted parameters. **e**–**g** Time courses of branch dissociation in the absence or presence of various concentrations of Coro7 construct β2CA (solid lines indicate double-exponential fits) and constructs β1β2 and CA (solid lines indicate single-exponential fits). The graphs in (**b**, **e**–**g**) show averages; each condition was repeated two to four times, with an average of 43 fields of view, 113 mother filaments, and 155 branches analyzed per condition. In (**c** and **d**), the error bars represent standard deviation (SD) of the best-fit parameters from the time courses shown in (**b**), as calculated using Origin software. Source data are provided in the Source Data file.

---

filament, β1 (or the β1-β2 linker) may also compete with D-loop binding to the hydrophobic cleft of Arp2. Thus, the proposed debranching mechanism is, in fact, a severing mechanism, analogous to how the six subdomains of gelsolin form a pincer around the actin filament during severing[46]. Gelsolin subdomain G1 (aided by subdomains G2 and G3) has been proposed to engage one side of the filament, bind in the hydrophobic cleft of an actin subunit and displace the D-loop of the subunit below, while subdomains G4-G6 perform a similar function on the opposite side of the filament[46,47]. By analogy, we propose that during debranching, Coro7 domains β2CA and β1 fulfill roles analogous to gelsolin subdomains G1-G3 and G4-G6, respectively.

In summary, we propose that while cortactin stabilizes the branch junction by strengthening the link between Arp2/3 complex and the daughter filament[9,48], Coro7 disassembles the branch junction by weakening this link. According to this model, the interface between Arp2/3 complex and the daughter filament serves as the preferred site for branch stabilization or destabilization by Arp2/3 complex regulators. This is because Arp2/3 complex readily binds to the pointed end[49], whereas its interaction with the mother filament is transient[50], unless stabilized by the daughter filament. Therefore, once Coro7 severs the link between Arp2/3 complex and the daughter filament, Arp2/3 complex immediately dissociates from the mother filament and returns to its inactive conformation (Supplementary Video 8). Consistently, GMF, which, like Coro7, promotes branch disassembly[14–16], also targets the interface between Arp2/3 complex and the daughter filament, albeit by a different mechanism, wedging into the hydrophobic cleft at the barbed end of Arp2[12].

## Methods
### Proteins
The cDNA of human Coro7 (UniProt: P57737-1) was obtained from DNASU (Arizona). Coro7 constructs (Fig. 1a) were PCR-amplified and cloned between the NotI (or NheI) and EcoRI sites of vector pMAL-c6T, resulting in MBP fusions with a TEV protease site inserted between MBP and the Coro7 constructs (primers used in this study as listed in Supplementary Table 1). MBP-CA was expressed in ArcticExpress (DE3) cells (Agilent). Transfected cells were cultured in Terrific Broth (TB) medium at 37 °C until an optical density of 1.5–2 at 600 nm was reached. Expression was induced with 0.5 mM isopropyl β-D-thiogalactoside (IPTG) and carried out for 20 h at 10 °C. Cells were harvested by centrifugation, resuspended in Amylose buffer [20 mM HEPES pH 7.5, 200 mM NaCl, 1 mM EDTA, 1 mM dithiothreitol (DTT)] supplemented with 100 mM phenylmethylsulfonyl fluoride (PMSF), and lysed using a microfluidizer (Microfluidics). The protein was purified on an amylose column using Amylose buffer, extensively washed in the same buffer, and eluted with 10 mM maltose. To remove MBP, TEV cleavage was performed at 4 °C overnight during dialysis against 20 mM HEPES pH 7.5, 200 mM NaCl, and 10 mM imidazole. Free MBP and TEV protease, both carrying a His-tag, were removed using a Ni-NTA affinity column (QIAGEN). The CA peptide was further purified by size exclusion chromatography on an SD75HL 16/60 column

(GE Healthcare), concentrated using a Vivaspin Turbo 3 kDa concentrator (Sartorius), and stored at −80 °C.

MBP-fused constructs FL, β1β2, and β2CA were subcloned between the SalI and EcoRI sites of vector pJC7 (AddGene)[51], and expressed in human Expi293F cells (Thermo Fisher Scientific) according to the manufacturer's protocol. Cells were harvested 72 h after transfection and stored at −80 °C. Pellets were thawed, resuspended in Amylose buffer supplemented with 0.5% Triton X-100, 1 mM PMSF, and a protease inhibitor cocktail (Sigma-Aldrich), and lysed using a Dounce homogenizer. Proteins were purified on an amylose column (NEB) as described above. Eluted MBP-fused proteins were diluted 4× to reduce the salt concentration and further purified on a MonoQ ion exchange column (Pharmacia) with a 50–500 mM NaCl gradient in 20 mM HEPES pH 7.5 and 1 mM EDTA. Pure proteins were concentrated, flash-frozen in liquid nitrogen, and stored at −80 °C.

Rabbit skeletal α-actin, mouse N-WASP WCA (residues 428–501), Arpin CA (residues 194–226), bovine brain Arp2/3 complex, and human Arp2/3 complex (carrying mutations L199C in Arp2 and L117C in Arp3) were expressed and purified as described[4,5,11].

For cell biology experiments, Coro7 constructs FL, β1β2, and β2CA were cloned into the custom-made plasmid MXS PGK Blasti bGHpA EF1 Flag iRFP670 Blue2 SV40 pA in place of Blue2 and between sites FseI and AscI. HEK293T cells were transfected with these plasmids using the calcium phosphate transfection method. For a 10 cm dish, 40 μg of plasmid DNA was mixed with 2 M CaCl₂ and 2× HBS buffer (55 mM HEPES pH 7, 0.27 M NaCl, 1.5 mM Na₂HPO₄·7H₂O). Cells were lysed 72 h post-transfection at 4 °C using XB-NP40 buffer (50 mM HEPES pH 7.7, 50 mM KCl, 1% NP-40) supplemented with protease inhibitors and rocked for 10 min. Cell extracts were clarified by two centrifugation steps at 1878×g for 10 min and incubated overnight at 4 °C with 500 μL anti-Flag-M2 resin (Sigma) equilibrated in XB-NP40 buffer. The resin was loaded into 20 mL columns, washed sequentially with XB-NP40 buffer and Protein buffer (50 mM HEPES pH 8, 150 mM KCl), and eluted in three rounds with 3× Flag peptide. The collected elutions were dialyzed against Protein buffer and concentrated using an Amicon Ultra-4 10 kDa MWCO. The purity and concentration of the proteins were estimated by running serial dilutions of the protein preparations alongside a bovine serum albumin (BSA) standard on Coomassie-stained gels.

### Actin polymerization assay
Time courses of actin polymerization were measured by monitoring the fluorescence increase resulting from the incorporation of pyrene-labeled actin into filaments, using a Cary Eclipse fluorescence spectrophotometer (Varian). Before data acquisition, 2 μM Mg-ATP-actin (6% pyrene-labeled) was mixed with 5 nM Arp2/3 complex, 200 nM N-WASP WCA, and the indicated concentrations of Coro7 constructs FL, β1β2, and CA, or Arpin CA in F-buffer (10 mM Tris-HCl pH 7.5, 1 mM MgCl₂, 50 mM KCl, 1 mM EGTA, 0.1 mM NaN₃, and 0.2 mM ATP) (Fig. 1c–e). The maximum polymerization rate was calculated using the equation $S' = (S \times M_t) / (f_{max} - f_{min})$, where $S'$ is the apparent

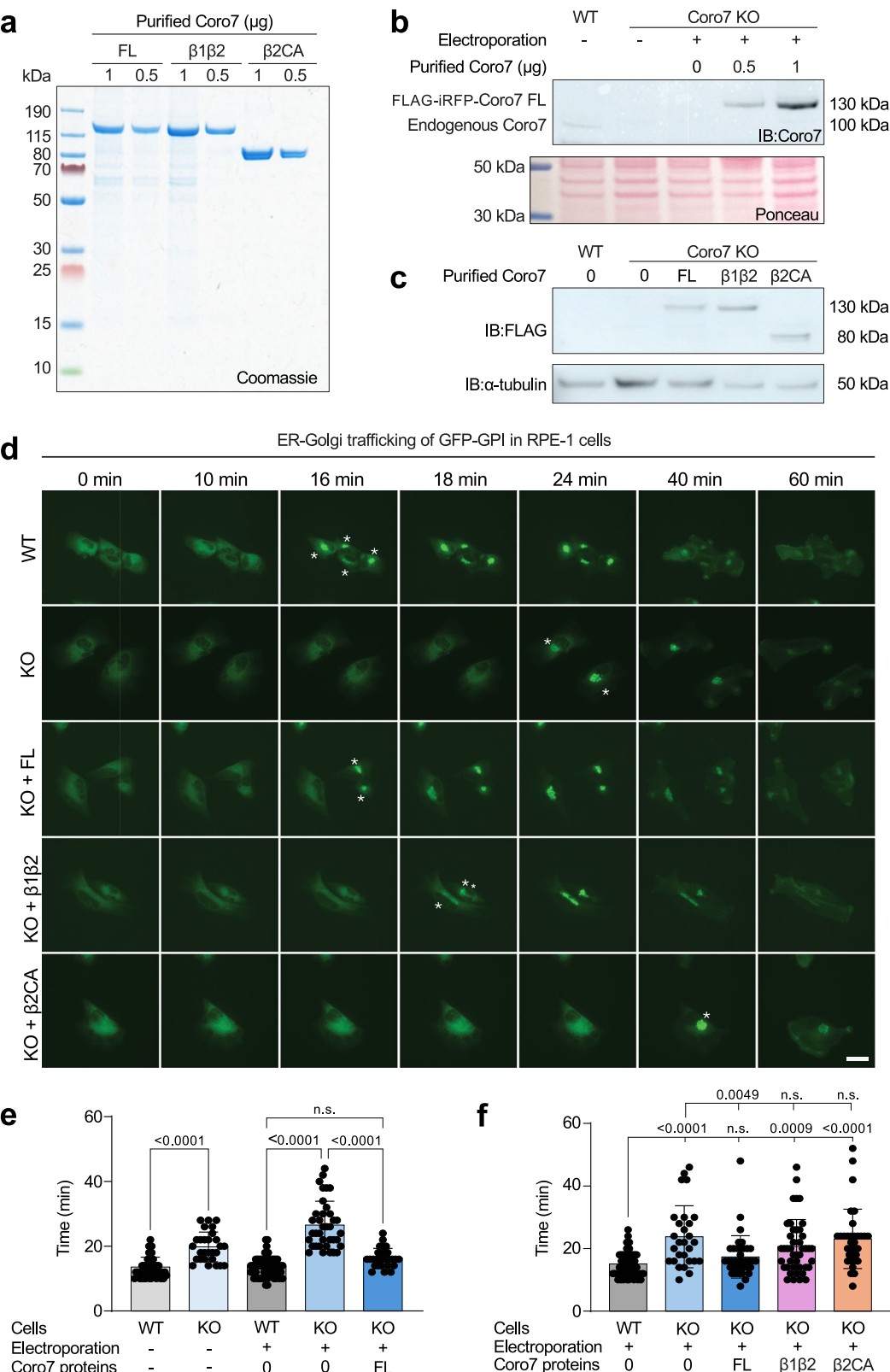

slope (μM s$^{-1}$), S is the maximum slope of the raw trace, M$_t$ is the concentration of polymerizable monomers, and f$_{min}$ and f$_{max}$ correspond to the fluorescence intensities at the start of the reaction and at the plateau, respectively. Rates are reported relative to the maximum polymerization rate of actin alone. Statistical significance was determined using an unpaired $t$-test with Welch's correction in Prism v7.0 ($p$-values reported in Fig. 1c–e).

**Isothermal titration calorimetry (ITC)**

ITC experiments were performed using a VP-ITC apparatus (MicroCal). Arp2/3 complex and Coro7 CA were dialyzed for two days against 20 mM HEPES pH 7.5, 100 mM KCl, 1 mM MgCl$_2$, 1 mM EGTA, and 1 mM DTT, supplemented with either 0.2 mM ATP or ADP. Titrations were conducted at 25 °C and consisted of 27 injections of 10 μL each (20 s per injection, with 300-s intervals

**Fig. 5 | Coro7 promotes ER-Golgi transport of GFP-GPI. a** Coomassie-stained SDS-PAGE of purified Flag-iRFP-tagged Coro7 FL, β1β2, and β2CA proteins. **b** Western blot of parental RPE-1 cells (WT), Coro7 KO cells, and KO cells electroporated with varying amounts of purified Coro7 FL, probed with Coro7 antibody. Endogenous Coro7 levels are most closely matched by electroporation of 0.5 µg Coro7 FL. Ponceau-stained membrane shown as a loading control. **c** Western blot of KO cell extracts electroporated with Coro7 proteins (0.5 µg Coro7 FL equivalent in each case), probed with anti-Flag M2 antibody. Anti-α-tubulin Western blot serves as a loading control. **d** GFP-GPI trafficking in RPE-1 cells using the RUSH system. Asterisks indicate the time at which the GFP-GPI signal condenses at the Golgi. Scale bar: 20 µm. **e** Quantification of GFP-GPI trafficking time from the ER (t = 0, biotin addition) to the Golgi (asterisks in part d). In WT cells, the average trafficking time was t = 13.80 ± 3.00 min (n = 62 cells), compared to t = 26.54 ± 7.37 min in KO cells (n = 37

cells), p < 0.0001. The KO phenotype was rescued by Coro7 FL electroporation (t = 16.43 ± 2.89 min, n = 28 cells), p < 0.0001. **f** Quantification of trafficking time in KO cells electroporated with Coro7 FL, β1β2, and β2CA, compared to WT (t = 15.10 ± 3.62 min, n = 62 cells) and KO (t = 23.79 ± 9.90 min, n = 28 cells) cells. Electroporation of Coro7 FL (t = 17.35 ± 6.77 min, n = 37 cells) rescued the KO phenotype, while β1β2 (t = 20.65 ± 8.64 min, n = 43 cells) and β2CA (t = 23.09 ± 9.48 min, n = 33 cells) did not (p = 0.7393 and p > 0.999, respectively). Data in (**e** and **f**) are pooled from three independent experiments (biological replicates). Data are presented as mean ± SD. Statistical significance was determined using one-way ANOVA followed by post hoc Sidak's multiple comparison tests (p-values reported in the figure; n.s., not significant, p > 0.05). Source data and statistical analysis are provided in the Source Data file.

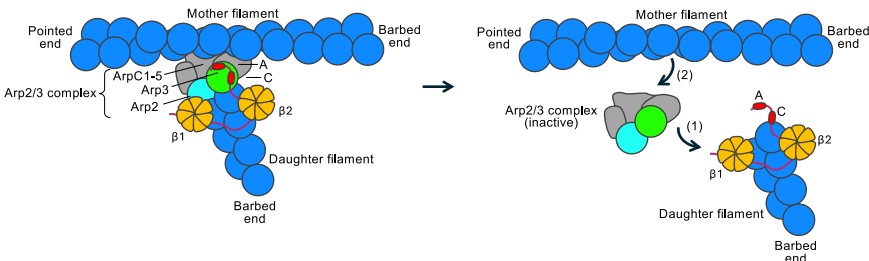

**Fig. 6 | Proposed model of branch disassembly by Coro7.** We propose that Coro7 A and β2 might bind first to exposed surfaces on Arp3 and the daughter filament, respectively. β2 is proposed to bind at the interface between the first two actin subunits along the long-pitch helix at the barbed end of Arp3. The long linker between β1 and β2 allows β1 to bind at multiple locations relative to β2. We favor the site at the interface between Arp2 and the actin subunit at its barbed end, as this would better explain Coro7's specificity for the branch junction. We hypothesize that debranching begins with the binding of Coro7 C's α-helix to the hydrophobic cleft of Arp3, which would competitively displace the D-loop of the actin subunit at

its barbed end. β1 (or the β1-β2 linker) may further assist in debranching by displacing the D-loop of the actin subunit at the barbed end of Arp2. Together, β1 on one side of the filament and β2CA on the opposite side form a pincer-like structure that severs the junction between Arp2/3 complex and the daughter filament (right, 1), similar to the mechanism by which gelsolin severs the actin filament[46]. Once the branch detaches, the affinity of Arp2/3 complex for the mother filament is low[50], leading to rapid dissociation and allowing the complex to revert to its inactive conformation (right, 2). See also Supplementary Video 8 and Supplementary Fig. 8.

between injections) of Coro7 CA (in the syringe) into Arp2/3 complex (in the 1.44 mL cell). The concentration of the proteins for each experiment is indicated in Fig. 2a. The heat of binding was corrected for the exothermic heat of injection, determined by injecting ligand into buffer. Data were analyzed using the Origin software (OriginLab Corporation).

## Cross-linking assay

As previously described[4,5,11], mutant human Arp2/3 complex (0.3 µM) was incubated with either 3 µM N-WASP WCA, 3 µM Coro7 CA, or 6 µM Arpin CA for 30 min at room temperature in KMEI buffer (50 mM KCl, 1 mM MgCl$_2$, 1 mM EGTA, 10 mM imidazole pH 7) supplemented with 0.2 mM ATP. BMOE (Thermo Fisher Scientific), freshly prepared in dimethyl sulfoxide, was added to a final concentration of 16 µM. Reactions were performed at 21 °C and quenched at the indicated time points (Fig. 2b) by adding an equal volume of 2× SDS-loading buffer (LI-COR Biosciences) supplemented with 100 mM β-mercaptoethanol (fresh). Samples were loaded onto 12% SDS-PAGE gels, transferred to PVDF membranes (Bio-Rad), and immunoblotted with an anti-Arp3 antibody (antibodies used in this study as listed in Supplementary Table 1). Membranes were imaged using a G-BOX scanner (Syngene) and analyzed densitometrically using Image Lab (Bio-Rad). Mean and SEM values were calculated from three or more independent experiments (Fig. 2b and Supplementary Fig. 1). A single-exponential fit was applied, and statistical significance was assessed using an unpaired t-test with Welch's correction in Prism v7.0 (p-values reported in Fig. 2b).

## Cryo-EM sample preparation, data acquisition, and processing

Bovine brain Arp2/3 complex (12 µM) was mixed with Coro7 CA (120 µM) in ATP-supplemented KMEI buffer. To improve particle

distribution, 1.6 mM CHAPSO was added before grid preparation. Cryo-EM grids were prepared by applying 3 µL of the sample onto glow-discharged (1 min, easiGlow, Pelco) R1.2/1.3 300-mesh Quantifoil holey carbon grids (Electron Microscopy Sciences). Grids were blotted for 2 s using Whatman 41 filter paper and flash-frozen by plunging into liquid ethane with a Vitrobot Mark IV (Thermo Fisher Scientific). The cryo-EM dataset was acquired using EPU software (Thermo Fisher Scientific) on a FEI Titan Krios transmission electron microscope operating at 300 kV, equipped with a Gatan K3 direct electron detector and an energy quantum filter. Images were collected in super-resolution mode at a nominal magnification of ×105,000, corresponding to a pixel size of 0.428 Å, with a defocus range of −0.5 to −2.5 µm.

Cryo-EM movies (7295) were imported into cryoSPARC v.4.4[52] and binned by two during patch motion correction, resulting in a working pixel size of 0.856 Å. Following patch contrast transfer function (CTF) estimation, micrographs with a CTF fit > 6 Å were excluded, leaving 7199 accepted micrographs. Approximately 1000 particles from 50 micrographs were manually picked and used to train the program Topaz[53] to produce an initial model for particle picking across the entire dataset. This was followed by reference-free 2D classification, and the Topaz model was re-trained with the selected particles. Particles picked with the second Topaz model (241121) were extracted with a box size of 320 pixels (274 Å) and binned by two to a box size of 160 pixels. Reference-free 2D classification was used to remove particles lacking structural features (32740), which were subjected to ab initio reconstruction to produce a "junk" volume for subsequent refinements. Accepted particles (208381) were used for ab initio reconstruction to produce a preliminary Arp2/3 complex volume. Five rounds of heterogeneous refinement were performed using the Arp2/3 complex and junk volumes as inputs. After each round, particles

placed into the junk class were discarded. The remaining, accepted particles (142687) were unbinned and subjected to two rounds of unfocused 3D classification without image alignment to eliminate poorly resolved particles. The resulting 116893 particles were subjected to global and local CTF refinement, reference-based motion correction, and non-uniform refinement to produce a consensus map at 2.97 Å resolution.

To improve density for poorly defined regions of the consensus map, 3D classification without image alignment was performed separately with masks focused on three regions: Arp2 subdomains 1-2, ArpC1 extension (residues I288–T321), and Arp3 D-loop (E39–V55). The resulting particle sets for these regions (59592, 23356, and 24356, respectively) were subjected to local refinement (Arp2 subdomains 1-2) or non-uniform refinement (ArpC1 extension and Arp3 D-loop), producing maps at resolutions of 3.07 Å, 3.25 Å, and 3.23 Å, respectively, in which these regions were better resolved. Other regions of the map were similarly improved using local refinements of the consensus map and masks focused on Arp3-ArpC3-Coro7 (3.06 Å), Arp3-ArpC2-Coro7 (2.99 Å), and ArpC1-ArpC4-ArpC5 (2.96 Å). Using the program ChimeraX[54], each map was fitted and resampled onto the consensus map. The best density for each region was extracted using command 'volume zone', and the densities were stitched together with command 'volume maximum' to produce a composite map used for model building and refinement. Supplementary Figures 2 and 3 illustrate the cryo-EM data processing workflow and map validation, respectively.

### Model building and refinement
Model building into the composite cryo-EM map was performed using the program Coot[55], starting from the cryo-EM structure of cortactin's acidic domain bound to Arp2/3 complex (PDB: 8TAH)[8]. The real-space refinement function of Phenix[56] was used to refine individual atomic positions and temperature factors. Final model quality and refinement statistics are provided in Table 1 and Supplementary Fig. 3. Because composite maps often exhibit an artificial bump in the map-to-model correlation plot at high spatial frequencies, leading to overestimated resolution, we report the FSC at a 0.5 threshold rather than the conventional 0.143 cutoff. Figures were prepared with the programs PyMOL (Schrödinger, LLC) and ChimeraX[54].

### F-actin cosedimentation
Rabbit skeletal α-actin (40 μM) was polymerized in ATP-supplemented KMEI buffer. Various Coro7 constructs (1.5 μM) were then incubated with increasing concentrations of F-actin (Supplementary Fig. 4 and Source Data file). Samples (100 μL) were incubated at 4 °C for 1 h and centrifuged at 278,000×$g$ for 30 min using a S100-AT3 rotor. Supernatants were mixed with 25 μL of 4× SDS-loading buffer. Pellets were washed with 100 μL of KMEI buffer, resuspended in 100 μL of fresh KMEI buffer, and mixed with 25 μL of 4× SDS-loading buffer. For experiments with ADP-BeF$_3^-$, 40 μM F-actin was pre-incubated for 15 min with 2 mM BeSO$_4$ and 10 mM NaF prior to mixing with Coro7 FL. The KMEI buffer used in this assay was similarly supplemented with 2 mM BeSO$_4$ and 10 mM NaF. Samples were analyzed by SDS-PAGE followed by Coomassie Blue staining, imaged using a G:BOX scanner (Syngene), and quantified densitometrically using Image Lab (Bio-Rad). Data were fitted to a binding curve using GraphPad Prism.

### Microfluidics-TIRF microscopy
Rabbit skeletal α-actin was labeled with biotin, Alexa 488 or Alexa 647[15]. Ca$^{2+}$-actin was converted to Mg$^{2+}$-actin by exchanging in 50 μM MgCl$_2$ and 200 μM EGTA, followed by incubation on ice for 5 min immediately before use[15]. Direct visualization of actin filaments and branches was performed using a Nikon Eclipse Ti2 total internal reflection fluorescence (TIRF) microscope equipped with an SR HP Apo TIRF ×100 objective (Nikon) and an Andor iXon897 EMCCD

camera. Multiple fields of view were captured at frame rates of 0.1 to 0.05 s$^{-1}$ and saved as N-dimensional (ND2) files for further analysis. Coverslips were cleaned and functionalized with 2–5% biotin-PEGylation[57]. Flow cells were prepared as described previously[15]. Buffers were introduced into the flow cell using a Fluigent Flow EZ™ microfluidic controller, and the applied force on branches was calculated based on the flow rate[15,58].

### Debranching experiments
Debranching experiments were performed in Polymerization buffer (10 mM Tris-HCl pH 7.5, 50 mM KCl, 1 mM MgCl$_2$, 1 mM EGTA, 0.2 mM ATP, 0.1 mM NaN$_3$, and 2 mM DTT) supplemented with 15 mM glucose, 0.02 mg/mL catalase, and 0.1 mg/mL glucose oxidase. Alexa 488-labeled actin filaments were polymerized through barbed end elongation of filament seeds (10 % biotinylated) immobilized on the coverslip via interaction with neutravidin[15]. A pre-equilibrated solution containing Arp2/3 complex, N-WASP WCA, and Alexa 488-labeled actin monomers was then introduced into the flow cell to allow branch formation. After 2 min, the coverslip was washed with Polymerization buffer supplemented with 0.1 μM unlabeled monomers[15] and Coro7 constructs were injected into the flow cell at a flow rate of 25 μL min$^{-1}$ and at the indicated concentrations (Fig. 4). Force was then applied by increasing the flow rate of Polymerization buffer to 200 μL min$^{-1}$. Data acquisition began before applying the force and continued for ~30 min after the force was applied. Debranching data were analyzed as previously described[15].

### Cell lines and Coro7 knockout
hTERT-immortalized RPE-1 human retinal pigment epithelial cells, stably expressing the Str-KDEL_SBP-EGFP-GPI plasmid for imaging GFP-GPI trafficking using the RUSH system, were provided by Gaëlle Boncompain and Franck Perez (Institut Curie, Paris). Cells were cultured in DMEM/F12 medium supplemented with 10% FBS, 100 U/mL penicillin/streptomycin, 5 μg/mL puromycin, and 10 U/L avidin. MCF10A cells (a gift from T. Dubois, Institut Curie) were cultured in DMEM/F12 medium (Gibco) supplemented with 5% horse serum (Sigma), 100 ng/mL cholera toxin (Sigma), 20 ng/mL epidermal growth factor (Sigma), 0.01 mg/mL insulin (Sigma), 500 ng/mL hydrocortisone (Sigma), and 100 U/mL penicillin/streptomycin (Gibco). HEK293T human kidney epithelial cells were maintained in DMEM medium supplemented with 10% FBS and 100 U/mL penicillin/streptomycin. Media and supplements were obtained from Life Technologies and Sigma-Aldrich. All cells were incubated at 37 °C in 5% CO$_2$.

A CRISPR/Cas9-based approach was used to generate a Coro7 knockout derivative from hTERT-immortalized RPE-1 cells, stably expressing the Str-KDEL_SBP-EGFP-GPI and from MCF10A cells. The sequences GTCCAAGTTCCGGCACACCG (exon 1) and CGAGCGCCA-GAGCGAGTCAC (intron 1) were cloned into the pRG2-GG vector and co-transfected with the pRG2-GG vector targeting the *ATP1A1* gene and the Cas9-expressing vector using Lipofectamine 2000 (Thermo Fisher Scientific). Transfected cells were selected with 0.5 μM ouabain for 10 days[59], and clones displaying biallelic deletions were identified by PCR on genomic DNA using the primers TTCAGGGTGTCCAAGTTCCG and CATCAGTCCTCCTGCCGTTT. The selected clone carries two deleted *CORO7* alleles, resulting in severely truncated Coro7 protein.

### Western blots
Cells were lysed in RIPA buffer supplemented with protease inhibitor (Roche). Lysates were clarified by centrifugation, and supernatants were analyzed by SDS-PAGE using NuPAGE 4–12% Bis-Tris gels (Life Technologies). Pieces of nitrocellulose membrane were cut, incubated with the indicated primary antibodies, HRP-conjugated secondary antibodies (Supplementary Table 1), and developed with SuperSignal™ West Femto Substrate (Thermo Fisher Scientific) and visualized using a Cytiva imaging system.

## Electroporation of purified Coro7 proteins

Purified Coro7 proteins (see above) were introduced into RPE-1 cells by electroporation using the Neon Transfection system (Invitrogen). Cells were resuspended in 10 μL of Protein buffer and gently mixed with the protein suspension. The mixture (10 μL) was aspirated using a Neon 10 μL pipette and electroporated with two pulses (1050 V, 30 milliseconds). Following electroporation, cells were resuspended in fresh media and incubated at 37 °C with 5% $CO_2$ for 5 h before live imaging or Western blot analysis.

## Live cell and video microscopy

RPE-1 GFP-GPI WT and Coro7 KO cells were used to assess synchronized trafficking from the ER to the Golgi with the RUSH system, a two-state assay that utilizes reversible streptavidin-based interactions. In this system, a minimal ER hook retains a labeled reporter protein, which is released upon incubation with biotin. Biotin binding to streptavidin disrupts the interaction and allows the reporter protein to traffic from the ER to the Golgi. RPE-1 GFP-GPI WT, Coro7 KO, and KO cells electroporated with Coro7 constructs were seeded onto glass-bottom μ-slides (ibidi). Before imaging, the culture medium was replaced with Fluorobrite DMEM supplemented with 10% FBS to minimize the background fluorescence. At time $t = 0$, biotin (Sigma-Aldrich) was added to a final concentration of 40 μM. Time-lapse imaging was performed at 37 °C in a thermostat-controlled chamber using a Zeiss epifluorescence microscope. Images were acquired with a ×63 objective at 2-min intervals for at least 1 h. Statistical analysis was conducted using GraphPad Prism software.

## Cortactin immunostaining and pVenus-GBF1 expression imaging in MCF10A cells

MCF10A parental and Coro7 KO cells were seeded on glass coverslips coated with 20 μg mL$^{-1}$ bovine fibronectin (Sigma) and transiently transfected with a plasmid encoding pVenus-GBF1 (a gift from C.L. Jackson, Institut Jacques Monod) using Lipofectamine 2000 (Invitrogen). Twenty-four hours post-transfection, cells were fixed with 4% paraformaldehyde, permeabilized with 0.5% Triton X-100, blocked with 1% BSA, and incubated for 1 h at room temperature with GFP-Booster ATTO488 (ChromoTek) and anti-cortactin antibody (Sigma 05-180-I) diluted 1:200. This was followed by a 1-h incubation with secondary antibodies at 5 μg/mL. Nuclei were counterstained with DAPI (Invitrogen).

Images were acquired using a Leica SP8 ST-WS confocal microscope equipped with an HC PL APO 63x/1.40 oil immersion objective, a white light laser, and HyD and PMT detectors. Acquisition settings were kept identical across all conditions. Image analysis was performed using ImageJ/FIJI software (https://imagej.net/). Mean cortactin intensity was measured in GBF1-positive cells by masking regions of interest corresponding to GBF1 localization sites, and values were normalized to total cellular intensity. Cells exhibiting unusually high pVenus-GBF1 fluorescence were excluded from the analysis. GraphPad Prism and Microsoft Excel were used for statistical analysis of data from three independent biological replicates. Normality tests were performed, and Mann-Whitney U nonparametric two-tailed tests were applied. A $p$-value < 0.05 was considered statistically significant.

## Reporting summary

Further information on research design is available in the Nature Portfolio Reporting Summary linked to this article.

## Data availability

Cryo-EM maps, models, and micrographs were deposited with the EMDB, PDB, and EMPIAR repositories. Accession codes are as follows: EMD-47408 (consensus map); EMD-47746, EMD-47769, EMD-47770, EMD-47772, EMD-47795 and EMD-47797 (locally refined and focused maps); EMD-47836 (composite map); PDB 9EAM (coordinates); and EMPIAR-12575 (micrographs). Source data are provided with this paper.

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

## Acknowledgements

This work was supported by the National Institutes of Health grants R01 GM073791 to R.D., R35 GM136656 to E.M.D.L.C., and Agence Nationale de la Recherche grants ANR-22-CE13-0041 and ANR-24-CE44-4957 to A.M.G. Data collection was performed at the National Cryo-EM Facility (NCEF), National Cancer Institute (NCI).

## Author contributions

All authors contributed to the conceptualization of the project. N.S.N., M.B., F.E.F., G.R., and A.J.S. conducted biochemical experiments. F.E.F., K.R.B., and R.D. acquired and processed the cryo-EM data and refined the structure. R.H. and A.M.G. carried out cellular experiments. All authors contributed to data analysis, figure preparation, and manuscript writing. R.D., E.M.D.L.C., and A.M.G. secured funding for the project.

## Competing interests

The authors declare no competing interests.
