## [Transparent Peer Review file · Nature Communications]

Mechanism of Arp2/3 Complex Branch Disassembly by Human Coro7

Corresponding Author: Professor Roberto Dominguez

Version 0:

Reviewer comments:

Reviewer #1

(Remarks to the Author)

This study investigates coronin 7 (Coro7), a class III coronin protein, and its contributions to ER-Golgi trafficking and actin network regulation. The authors demonstrate that full-length Coro7 (FL) is necessary for proper localization and function in Golgi trafficking, while truncated constructs exhibit distinct effects on Arp2/3-mediated actin polymerization. Biochemical assays, cryo-EM, and TIRF experiments reveal that only FL Coro7 acts as an effective debrancher, whereas the CA domain alone inhibits Arp2/3 polymerization. The cryo-EM structure of the CA domain bound to Arp2/3 provides mechanistic insight, though its physiological relevance remains unclear given that this domain may be occluded in the full-length protein.

Overall, the experiments are rigorously performed, the manuscript is well-written and employs a robust combination of techniques to dissect Coro7's mechanisms. However, some methodological and interpretive concerns should be addressed before publication. I recommend acceptance with revisions.

Minor Comments

1. The manuscript mentions Coro7's F-actin binding sites, yet these are not depicted in the domain diagram (Figure 1). I think clarifying their locations would help readers better understand Coro7's functional architecture.
2. The binding affinity of Coro7 for actin is not mentioned/measured. Given its relevance to the TIRF experiments, I recommend either quantifying it or citing a previous study (if its been measured before) – I think readers will find this helpful.
3. In the TIRF experiments, 0.2 μ M unlabeled actin is being flowed alongside Coro7 at 200 μ L/min flow. Even though the growth will be short, the barbed ends will grow at this concentration – bringing force into the picture. How is daughter filament growth controlled? My concern is that invisible actin incorporation could influence premature detachment as elongation (however slow) will lead to longer filaments (with time-dependent changes in length), which in turn will translate non-constant force.
4. Was a no-flow control performed to rule out mechanical artifacts?
5. I recommend stating the number of repeats and filaments analyzed. Are the graphs representative or averaged across replicates?
6. While the CA-Arp2/3 structure is elegant, it may not reflect physiological conditions if this domain is buried in FL Coro7. Did the authors attempt FL Coro7 + Arp2/3 (with/without actin) to resolve this?
7. The CA domain's inhibition of Arp2/3 (Figure 2) contrasts with FL Coro7's debranching. Is FL Coro7's activity nucleotide-dependent, like CA?
8. The discussion (Paragraph 3) claims all domains are needed for dissociation, but only truncations were tested. Are structural motifs or specific sequences critical? Mutational studies (e.g., key residue swaps) could clarify this.
9. Figure 1b is not mentioned in the text. Either reference it or remove it.
10. Figure 3 (Coro7 CA Binding Section): The construct naming is confusing (CA, C, A, "Coro7"). I recommend specifying "full-length Coro7" where applicable or clarify which truncation is being discussed.
11. Discussion, paragraph 7: The claim that "Coro7 remains bound to either Arp2/3 or actin after debranching" needs clarification. Is this observed or hypothesized? If observed, provide evidence; if hypothesized, state so explicitly.
12. Final paragraph: Correct to: "Consistently, GMF, which is like Coro7, promotes branch disassembly, also..."

Reviewer #2

(Remarks to the Author)

Coronins constitute a family of conserved proteins that play various roles in actin cytoskeleton dynamics. Among the seven coronin proteins found in humans, Coro7 is the most divergent (it contains 2 beta-propellers instead of one), while it also remains poorly characterized compared to the other coronins. The manuscript presented by Nejad et al. aims to close this gap by characterizing Coro7 using biochemical, structural and cell-biological approaches, mostly with respect to its interaction with the Arp2/3 complex.

This study provides insights into Coro7 function and will therefore be valuable for the actin community. However, several parts of the manuscript feel incomplete, and the final debranching model is too speculative. The following points will need to be addressed before the manuscript is suitable for publication:

1. One important feature of coronins is their ability to bind to F-actin. Unfortunately, the manuscript only briefly touches on this, describing one experiment (Supplementary Fig. 4) to characterize the interaction. Yet, the proposed debranching model shown in Fig. 6 and the supplementary video heavily depends on the sequential binding of both beta propellers of Coro7 to F-actin.

The co-sedimentation experiment in Supplementary Fig. 4 revealed that only a minor fraction of Coro7 pellets with F-actin, suggesting a weak interaction. However, the experiments are only performed with the truncated B1B2 construct, and not with full-length Coro7. In addition, it remains unclear whether the individual B1 and B2 propellers are capable of binding F-actin, even though this is assumed in the model. It could very well be that only B1 and not B2 has affinity for F-actin, or vice versa.

Therefore, to support their model, these co-sedimentation assays should be repeated with different Coro7 domains. Otherwise, the presented debranching model is much too speculative.

2. Continuing on the interaction of Coro7 with F-actin. Is the final structural model of Coro7 bound to the F-actin-Arp2/3 complex derived from an alpha-fold model? It is probably a bit misleading to show it like this without empirical data supporting it.

The Bear lab has previously shown that another coronin family member, Coro1B, displays higher affinity for F-actin in the ATP/ADP-Pi nucleotide state than in the ADP state (PMID: 17456547). Have the authors investigated this for Coro7? It might explain the weak and incomplete binding of Coro7 to F-actin in their assay, and using Coro7 in the ATP/ADP-Pi state might stabilize the interaction with actin and even enable characterization of the complex of full-length Coro7 with F-actin using cryo-EM.

3. The cryo-EM data are of high quality. However, in Supplementary Fig. 3d, the map to model FSC curve looks highly unusual. The curve shows an extra bump at high spatial frequency. Based on the "Model building and refinement" method section, it is ambiguous whether the final Phenix refinement was performed using the consensus map or the composite map. Refinements in composite maps remain a point of discussion within the cryo-EM field, because these maps are artificially constructed. If not done so, I would recommend the authors to refine their model in the consensus map, or at least write very clearly how the model was refined.

4. Continuing on the cryo-EM data: the authors have performed an extensive masked classification strategy to obtain the best possible density for each domain of the complex. This suggests that the complex displays flexibility. Out of curiosity, have the authors attempted to characterize this flexibility, e.g., using 3D Flexible Refinement in CryoSPARC? This could potentially reveal additional insights into the debranching mechanism.

5. For the debranching experiments with microfluidics-TIRF microscopy, the main figure (Fig. 4) only depicts the results and not the experimental data. The supplementary videos nicely show the data, but I would advice to also add some experimental images to Fig. 4. This will greatly benefit the reader to directly assess which type of experiments are analyzed.

6. The cellular assays are interesting, but it remains difficult to assess whether the observed effects on Golgi-trafficking are a direct result of changes in network debranching. For instance, Coro7 also interacts with other Golgi-proteins such as Cdc42 (PMID 27143109). I understand it is difficult to directly check for debranching, but it should be explicitly stated that causality between Golgi-transport and debranching rates cannot be determined.

Minor points.

7. Line numbers would have been very useful.

8. Fig. 3a – the local-resolution colored map is not very useful here – it can be moved to the supplement. Instead, I would show the cryo-EM map in a different orientation, to show the CA binding site from a different angle.

9. Page 3: these four regulators likely to compete these four regulators are likely to compete

10. Is the MBP-tag on the Coro7 constructs required for stability? If not, it may be useful to remove it to exclude artifacts from the large tag in e.g., the co-sedimentation assays.

Reviewer #3

(Remarks to the Author)
Nejad et al

In this manuscript, the authors use a combination of structural, biochemical and cell biological approaches to study the function of the coronin family member, *coro7*. This particular coronin represents a unique class of coronins with two beta propellers and an acid domain with noted similarity to the acidic domains found in many NPFs of the WASP family. They solve the structure of this acidic domain bound to the Arp2/3 complex which is a novel contribution to the literature. In addition, they go on to show that this protein has Arp2/3-debranching activity, similar to other coronins. Finally, they show that *coro7* knockout has defects in ER to Golgi trafficking using a chemical genetic approach (RUSH assay). Overall, this manuscript advances our understanding of *coro7* structure and function. Here, I offer a few critiques for the authors to consider in revising their manuscript:

1. The authors use a structure-function approach to both their in vitro debranching and nucleation assays, as well as in their cell biology work. They use a version of *coro7* lacking the acidic domain as representing the 'F-actin binding site'. However, they never show that this domain, which constitutes something like 90% of the protein actually binds F-actin. Rather they cite some quite old papers where this F-actin binding was never very well characterized. They've done very nice job of documenting the Arp2/3 interaction and should do the same for F-actin binding since their proposed mechanism in the pretty movie depends on it. Ideally, they would identify point mutations that block or diminish F-actin binding in one or both of the beta propellers.

2. The acidic domain is claimed to be part of a CA domain, but the 'C' part of this region is not very well defined. It would be useful to graphically illustrate the entire domain with some indication of how conserved it is. Another puzzling thing about this domain is that Xie et al (PNAS 2021) showed that the acidic domain is NOT required for debranching, in contrast to this work. It is ok to have contradictory data, but it is not ok to simply ignore those results. Just discuss possible explanations for the discrepancy. This helps the field.

3. The role of *coro7* in ER to Golgi trafficking that they claim as their main evidence for physiological function should be characterized more thoroughly. For example, does the *coro7* KO have overly stable patches of Arp2/3-branched actin on the ER and/or Golgi? Is this assay actually perturbed by inhibition of Arp2/3 function (e.g. treatment with CK666)? Since the F-actin binding is not very well defined (see point #1 above), we don't really know that this function depends on *coro7*'s ability to bind F-actin. For example, *coro7* binds to EPS15 and drives F-actin assembly on Golgi as well (see Yuan, Mol Cell, 2014), so their phenotype may have nothing to do with Arp2/3-debranching.

Version 1:

Reviewer comments:

Reviewer #1

(Remarks to the Author)

The authors have addressed all my concerns and I am happy to recommend acceptance of this manuscript.

Reviewer #2

(Remarks to the Author)

The authors have addressed most of my concerns and the manuscript is almost ready for publication. One remaining thing: even with the new co-sedimentation assays, I still find the debranching model shown in Supplementary Video 8 highly speculative in the absence of any direct structural data. Statements about which propeller "likely binds first to the filament" are too strong, and should be replaced with "might bind first to the filament". Structures will be required to obtain a more reliable model – I am happy to read in the rebuttal that the authors are pursuing this.

Finally, I may have missed it, but I also could not find titles and legends for all supplementary videos.

Reviewer #3

(Remarks to the Author)

The authors have addressed my criticism for the most part. I would still like to see them address the discrepancies between their findings and Xie et al (PNAS). This does not have to be particularly aggressive or confrontational statement, but just to acknowledge that they got different results and provide a possible explanation of why. This is basic scholarship 101.

We thank the reviewers for their constructive comments, which prompted us to obtain new data and provide clarifications that, we believe, have strengthened the manuscript.

Major changes to the Main Text and Supplementary Information files are highlighted in yellow.

Major figure revisions and newly added figures include:

- **Fig. 1a:** Boundaries of the C and A regions added
- **Fig. 3a (right):** New panel added
- **Fig. 4a:** Added and figure reformatted
- **Supplementary Fig. 3e:** Added (moved from main text)
- **Supplementary Fig. 4:** Added
- **Supplementary Fig. 5a:** Added
- **Supplementary Fig. 7:** Added

Reviewer 1

This study investigates coronin 7 (Coro7), a class III coronin protein, and its contributions to ER-Golgi trafficking and actin network regulation. The authors demonstrate that full-length Coro7 (FL) is necessary for proper localization and function in Golgi trafficking, while truncated constructs exhibit distinct effects on Arp2/3-mediated actin polymerization. Biochemical assays, cryo-EM, and TIRF experiments reveal that only FL Coro7 acts as an effective debrancher, whereas the CA domain alone inhibits Arp2/3 polymerization. The cryo-EM structure of the CA domain bound to Arp2/3 provides mechanistic insight, though its physiological relevance remains unclear given that this domain may be occluded in the full-length protein.

Overall, the experiments are rigorously performed, the manuscript is well-written and employs a robust combination of techniques to dissect Coro7's mechanisms. However, some methodological and interpretive concerns should be addressed before publication. I recommend acceptance with revisions.

Minor Comments

1. The manuscript mentions Coro7's F-actin binding sites, yet these are not depicted in the domain diagram (Figure 1). I think clarifying their locations would help readers better understand Coro7's functional architecture.

We do not yet know the specific amino acids in Coro7 that mediate actin binding—only that the two β -propeller domains, which together comprise essentially the entire protein, are involved. Therefore, adding putative binding sites to the domain map in Fig. 1a would do little to clarify the interaction, especially since the β -propellers may bind at the interface between F-actin and Arp2/3 complex. The model in Supplementary Fig. 8 further develops this idea, but we lack sufficient information to reliably identify specific amino acids involved in F-actin binding.

2. The binding affinity of Coro7 for actin is not mentioned/measured. Given its relevance to the TIRF experiments, I recommend either quantifying it or citing a previous study (if its been measured before) – I think readers will find this helpful.

We have now added new experimental evidence showing that Coro7 FL and the individual β -propeller domains bind F-actin with apparent affinities in the 5–11 μ M range (Supplementary Fig. 4a). However, we caution that these values are approximate and should not be overstated, given the limited resolution of the co-sedimentation assay. The results also suggest that the actin-binding sites may not be fully accessible in Coro7 FL and may become exposed only upon binding to the branch junction—that is, in the presence of Arp2/3 complex and both mother and daughter filaments. Additionally, we show that

Coro7 binds with lower affinity to ADP-BeF₃-F-actin than to ADP-F-actin (Supplementary Fig. 4b), consistent with a preference for older branches.

3. In the TIRF experiments, 0.2 μM unlabeled actin is being flowed alongside Coro7 at 200 $\mu\text{L}/\text{min}$ flow. Even though the growth will be short, the barbed ends will grow at this concentration – bringing force into the picture. How is daughter filament growth controlled? My concern is that invisible actin incorporation could influence premature detachment as elongation (however slow) will lead to longer filaments (with time-dependent changes in length), which in turn will translate non-constant force.

This is an important consideration that we have addressed previously (Pandit et al., PMC7306818), and we have now added clarifications in the current manuscript. First, we note that the concentration of unlabeled actin was erroneously listed as 0.2 μM ; the actual concentration was 0.1 μM , corresponding to the critical concentration of the barbed-end (Pollard TD, PMC2114620). Given an association rate constant (k^+) of 10 $\mu\text{M}^{-1}\text{s}^{-1}$ and a dissociation rate constant (k^-) of 1 s^{-1} , G-actin at 0.1 μM should add to and dissociate from filament ends at a rate of ~ 1 monomer per second, resulting in minimal net growth—especially over the short time window required for Coro7-mediated debranching. Indeed, in our experiments, the average branch length was $2 \pm 0.5 \mu\text{m}$ at the start of debranching, corresponding to a force of $0.45 \pm 0.11 \text{ pN}$ at 200 $\mu\text{L}/\text{min}$ flow. Coro7-dependent debranching is completed within 2 minutes (fast phase, $t_{1/2} \sim 20$ seconds). We directly measured filament length and found it increases by at most 0.15 μm during this period, resulting in a negligible effect on the force exerted on branches.

4. Was a no-flow control performed to rule out mechanical artifacts?

Yes, control experiments were conducted \pm Coro7 at a very low flow rate (25 $\mu\text{L}/\text{min}$), corresponding to an estimated force of 0.05 pN. In response to this question, we have added two new figures, Fig. 4a and Supplementary Fig. 5a, showing the difference in debranching as a function of flow (200 vs. 25 $\mu\text{L}/\text{min}$). As shown in Supplementary Fig. 5a, at 25 $\mu\text{L}/\text{min}$, branches remain after 13 minutes in the absence of Coro7, whereas the presence of 70 nM Coro7 removes most branches.

5. I recommend stating the number of repeats and filaments analyzed. Are the graphs representative or averaged across replicates?

As requested, this has now been added to the figure legend. The graphs show averages; each condition was repeated two to four times, with an average of 43 fields of view, 113 mother filaments, and 155 branches analyzed per condition.

6. While the CA-Arp2/3 structure is elegant, it may not reflect physiological conditions if this domain is buried in FL Coro7. Did the authors attempt FL Coro7 + Arp2/3 (with/without actin) to resolve this?

Indeed, the current structure reveals how Coro7 CA binds inactive Arp2/3 complex—crucial information, albeit incomplete. We fully agree that a desirable goal is to determine the structure of Coro7 FL at the branch junction, a challenging task currently underway. While we have already resolved the branch at 2.7- \AA resolution, most branches lack Coro7 because, upon binding, it debranches immediately—a common limitation structural biologists face with transient interactions. To increase the number of Coro7-bound branches, we are therefore testing mutants designed to slow debranching. So yes, it's coming!

7. The CA domain's inhibition of Arp2/3 (Figure 2) contrasts with FL Coro7's debranching. Is FL Coro7's activity nucleotide-dependent, like CA?

There may have been a misunderstanding. By ITC, the affinity of CA for Arp2/3 complex alone is very similar in the ADP and ATP states (Fig. 2a). Additionally, we have now examined binding of Coro7 FL to F-actin in the ADP and ADP-BeF₃ states (Supplementary Fig. 4). We found that Coro7 binds ADP-F-actin with higher affinity than ADP-BeF₃-F-actin, consistent with a preference for older branches. This contrasts with results obtained with Coro1B, shown to have higher affinity for F-actin in the ADP-Pi state than in the ADP state (PMID: 17456547). Although we do not want to belabor this difference, as we place limited confidence in cosedimentation assays, it would not make sense for cells to form a branch and destroy it immediately afterward—before Pi release. Consistently, we previously found that another debrancher, GMF, also binds preferentially to ADP-Arp2/3 complex (PMC3764776), and in that case we were able to carry out the experiments by ITC, a more reliable method.

8. The discussion (Paragraph 3) claims all domains are needed for dissociation, but only truncations were tested. Are structural motifs or specific sequences critical? Mutational studies (e.g., key residue swaps) could clarify this.

We are treating Coro7 as the sum of three domains— β -propellers 1 and 2, and CA—which led us to test six constructs (FL, β 1 β 2, β 1, β 2, β 2CA, and CA). At present, we lack information about specific amino acids involved in interactions at the branch junction (aside from CA) that could guide targeted mutational analysis. As mentioned above, work on the branch-bound structure is ongoing and will likely inform future efforts in this direction.

9. Figure 1b is not mentioned in the text. Either reference it or remove it.

Fig. 1b is referenced in the first paragraph of the Results section.

10. Figure 3 (Coro7 CA Binding Section): The construct naming is confusing (CA, C, A, "Coro7"). I recommend specifying "full-length Coro7" where applicable or clarify which truncation is being discussed.

Indeed, this was confusing. We now refer to the fragments as Coro7 CA, Coro7 C, and Coro7 A, and to the full-length protein as Coro7 FL.

11. Discussion, paragraph 7: The claim that "Coro7 remains bound to either Arp2/3 or actin after debranching" needs clarification. Is this observed or hypothesized? If observed, provide evidence; if hypothesized, state so explicitly.

This is not an observation but a hypothesis, based on the debranching mechanism proposed in the Results and Discussion, including newly added clarifications. However, we do not know how long Coro7 remains bound to Arp2/3 complex, actin, or both after debranching. Accordingly, as suggested, we have now qualified this hypothesis in the Discussion.

12. Final paragraph: Correct to: "Consistently, GMF, which is like Coro7, promotes branch disassembly, also..."

Have we misunderstood this comment? GMF is not like Coro7 in structure or mechanism, but, like Coro7, it promotes branch disassembly. Thus, the original sentence is correct: "Consistently, GMF, which like Coro7 promotes branch disassembly, also targets the interface between Arp2/3 complex and the daughter filament, albeit by a different mechanism, wedging into the hydrophobic cleft at the barbed end of Arp2"

Reviewer 2

Coronins constitute a family of conserved proteins that play various roles in actin cytoskeleton dynamics. Among the seven coronin proteins found in humans, Coro7 is the most divergent (it contains 2 beta-propellers instead of one), while it also remains poorly characterized compared to the other coronins. The manuscript presented by Nejad et al. aims to close this gap by characterizing Coro7 using biochemical, structural and cell-biological approaches, mostly with respect to its interaction with the Arp2/3 complex.

This study provides insights into Coro7 function and will therefore be valuable for the actin community. However, several parts of the manuscript feel incomplete, and the final debranching model is too speculative. The following points will need to be addressed before the manuscript is suitable for publication:

1. One important feature of coronins is their ability to bind to F-actin. Unfortunately, the manuscript only briefly touches on this, describing one experiment (Supplementary Fig. 4) to characterize the interaction. Yet, the proposed debranching model shown in Fig. 6 and the supplementary video heavily depends on the sequential binding of both beta propellers of Coro7 to F-actin.

The co-sedimentation experiment in in Supplementary Fig. 4 revealed that only a minor fraction of Coro7 pellets with F-actin, suggesting a weak interaction. However, the experiments are only performed with the truncated B1B2 construct, and not with full-length Coro7. In addition, it remains unclear whether the individual B1 and B2 propellers are capable of binding F-actin, even though this is assumed in the model. It could very well be that only B1 and not B2 has affinity for F-actin, or vice versa.

Therefore, to support their model, these co-sedimentation assays should be repeated with different Coro7 domains. Otherwise, the presented debranching model is much too speculative.

We have now added new experimental evidence showing that FL Coro7 and the β -propeller domains—both together and independently—cosediment with F-actin (Supplementary Fig. 4). The apparent affinities for all constructs analyzed fall within the 5–11 micromolar range. This was unexpected, as we did not observe an additive effect for $\beta 1\beta 2$ compared to the individual β -propeller domains, suggesting that, like CA, the F-actin binding surface is at least partially buried within Coro7 FL. Our working model is that Coro7 opens at the branch junction, with the β -propeller domains and CA region cooperating to specifically target the junction. This interpretation appears mechanistically sensible, as tight, nonspecific binding to either F-actin or Arp2/3 complex would likely result in unproductive interactions. This point is now addressed in the Discussion. We also caution that affinity values derived from cosedimentation assays should be interpreted with care.

2. Continuing on the interaction of Coro7 with F-actin. Is the final structural model of Coro7 bound to the F-actin-Arp2/3 complex derived from an alpha-fold model? It is probably a bit misleading to show it like this without empirical data supporting it.

The Bear lab has previously shown that another coronin family member, Coro1B, displays higher affinity for F-actin in the ATP/ADP-Pi nucleotide state than in the ADP state (PMID: 17456547). Have the authors investigated this for Coro7? It might explain the weak and incomplete binding of Coro7 to F-actin in their assay, and using Coro7 in the ATP/ADP-Pi state might stabilize the interaction with actin and even enable characterization of the complex of full-length Coro7 with F-actin using cryo-EM.

AlphaFold was used, among other sources, to generate the model shown in Supplementary Fig. 8, which suggested the most likely position of the β -propellers on F-actin. Other sources of information included the structure of Arp2/3 complex with bound Coro7 CA determined here, and Emil Raisler's low-

resolution cryo-EM map (8.6 Å) of the single β-propeller domain of yeast coronin-1 bound to F-actin. Coincidentally, the AlphaFold model is mostly in agreement with this map, providing additional confidence in the model. However, because binding of the β-propellers to F-actin is not the same as binding of Coro7 FL to the branch, we also used general structural logic, such as limitations imposed by the length of the linkers and some parallels with known mechanisms of GMF branch disassembly and cortactin branch stabilization. Nevertheless, we cannot stress enough that this is just a hypothetical model. We have reworded the parts of the Discussion describing this model to remove any potential ambiguity.

Binding of Coro7 (FL and domains) to F-actin has now been added (Supplementary Fig. 4a). As detailed in our response to Reviewer 1 (point 7), we also found that Coro7 FL binds with lower affinity to F-actin in the ADP-BeF₃ state than in the ADP state (Supplementary Fig. 4b), consistent with a preference for older branches and in contrast to the results reported for Coro1B.

3. The cryo-EM data are of high quality. However, in supplementary Fig. 3d, the map to model FSC curve looks highly unusual. The curve shows an extra bump at high spatial frequency. Based on the “Model building and refinement” method section, it is ambiguous whether the final Phenix refinement was performed using the consensus map or the composite map. Refinements in composite maps remain a point of discussion within the cryo-EM field, because these maps are artificially constructed. If not done so, I would recommend the authors to refine their model in the consensus map, or at least write very clearly how the model was refined.

Very good point—this is now clarified in the Methods section. A composite map was indeed used for model building and refinement. As the reviewer correctly notes, this explains the bump at high spatial frequencies, which is commonly observed when fitting a model to a composite map. To account for this, we report the map-to-model resolution using the 0.5 FSC threshold rather than the conventional 0.143 cutoff, as the bump would otherwise lead to an artificially inflated resolution estimate.

4. Continuing on the cryo-EM data: the authors have performed an extensive masked classification strategy to obtain the best possible density for each domain of the complex. This suggests that the complex displays flexibility. Out of curiosity, have the authors attempted to characterize this flexibility, e.g., using 3D Flexible Refinement in CryoSPARC? This could potentially reveal additional insights into the debranching mechanism.

We did try 3D variability analysis (not 3D Flex) on the consensus map particles, but unlike in other structures we have determined (e.g., PMC10434305), nothing stood out.

5. For the debranching experiments with microfluidics-TIRF microscopy, the main figure (Fig. 4) only depicts the results and not the experimental data. The supplementary videos nicely show the data, but I would advice to also add some experimental images to Fig. 4. This will greatly benefit the reader to directly assess which type of experiments are analyzed.

We like this suggestion, and in response to it and a related comment from Reviewer 1, we have now added two new figures, Fig. 4a and Supplementary Fig. 4b and Supplementary Videos 2-6, showing examples of debranching experiments as a function of flow (200 vs. 25 μL/min), time, and the presence/absence of Coro7 constructs.

6. The cellular assays are interesting, but it remains difficult to assess whether the observed effects on Golgi-trafficking are a direct result of changes in network debranching. For instance, Coro7 also interacts

with other Golgi-proteins such as Cdc42 (PMID 27143109). I understand it is difficult to directly check for debranching, but it should be explicitly stated that causality between Golgi-transport and debranching rates cannot be determined.

To circumvent technical difficulties in specifically linking the Coro7 KO phenotype to debranching during ER-to-Golgi transport, we imaged branched actin networks associated with vesicles enriched in the perinuclear region of MCF10A cells, another cell line for which we generated a Coro7 knockout. Imaging was more robust for cortactin than for Arp2/3 complex subunits, likely because cortactin is a bona fide branch marker and more accessible to antibodies, whereas Arp2/3 complex subunits can be partially buried at the branch junction. Consistent with Coro7 promoting debranching of Arp2/3 complex networks, we observed increased cortactin staining in Coro7 KO cells, which colocalized with GBF1 at the ERGIC. This new data is now included as Supplementary Fig. 7.

Minor points

7. Line numbers would have been very useful.

This may have occurred during the manuscript reformatting by NC, as line numbers were present in our submitted 'MS_and_Figures.pdf' file. We will monitor this carefully during resubmission.

8. Fig. 3a – the local-resolution colored map is not very useful here – it can be moved to the supplement. Instead, I would show the cryo-EM map in a different orientation, to show the CA binding site from a different angle.

We moved the local-resolution colored map to Supplementary Fig. 3e and now show, in its place, another view rotated by 90°.

9. Page 3: *these four regulators likely to compete* – these four regulators are likely to compete

Corrected, thank you

10. Is the MBP-tag on the Coro7 constructs required for stability? If not, it may be useful to remove it to exclude artifacts from the large tag in e.g., the co-sedimentation assays.

The MBP-tag is indeed necessary for stability

Reviewer 3

Nejad et al.

In this manuscript, the authors use a combination of structural, biochemical and cell biological approaches to study the function of the coronin family member, Coro7. This particular coronin represents a unique class of coronins with two beta propellers and an acidic domain with noted similarity to the acidic domains found in NPFs of the WASP family. They solved the structure of this acidic domain bound to the

Arp2/3 complex which is a novel contribution to the literature. In addition, they go on to show that this protein has Arp2/3-debranching activity, similar to other coronin families. Finally, they show that Coro7 knockout has defects in ER to Golgi trafficking using a chemical genetic approach (RUSH assay). Overall, this manuscript advances our understanding of Coro7 structure and function. Here, I offer a few critiques for the authors to consider in revising their manuscript:

1. The authors use a structure-function approach to both their in vitro debranching and nucleation assays, as well as in their cell biology work. They use a version of Coro7 lacking the acidic domain as representing the 'F-actin binding site'. However, they never show that this domain, which constitutes something like 90% of the protein actually binds F-actin. Rather they cite some quite old papers where this F-actin binding was never very well characterized. They've done very nice job of documenting the Arp2/3 interaction and should do the same for F-actin binding since their proposed mechanism in the pretty movie depends on it. Ideally, they would identify point mutations that block or diminish F-actin binding in one or both of the beta propellers.

We thank the reviewer for the positive feedback. We note that while our original submission already included F-actin cosedimentation data for Coro7 construct $\beta 1\beta 2$, all three reviewers commented on the lack of additional cosedimentation data. In response to these comments, we have now added Supplementary Fig. 4a, showing cosedimentation data for four Coro7 constructs: FL, $\beta 1$, $\beta 2$, and $\beta 1\beta 2$. An intriguing observation was that the apparent binding affinities of all constructs were similar, falling in the 5–11 micromolar range. In other words, the affinity of $\beta 1\beta 2$ (or FL) did not appear greater than that of $\beta 1$ or $\beta 2$ alone, suggesting that the F-actin binding surfaces on both β -propeller domains are at least partially occluded within the FL protein. This is consistent with our observation that CA, which inhibits Arp2/3 complex in isolation, fails to do so as part of Coro7 FL (Fig. 1c,e). We caution the reader that affinities derived from cosedimentation should not be overstated, as they are not as reliable as those obtained from ITC (e.g. Fig. 2a). We also show that Coro7 binds with lower affinity to ADP-BeF₃-F-actin than to ADP-F-actin (Supplementary Fig. 4b), consistent with a preference for older branches.

2. The acidic domain is claimed to be part of a CA domain, but the 'C' part of this region is not very well defined. It would be useful to graphically illustrate the entire domain with some indication of how conserved it is. Another puzzling thing about this domain is that Xie et al (PNAS 2021) showed that the acidic domain is NOT required for debranching, in contrast to this work. It is ok to have contradictory data, but it is not ok to simply ignore those results. Just discuss possible explanations for the discrepancy. This helps the field.

The reviewer is correct—the only thing clearly conserved about the C regions of Coro7, NPFs, and Arpin is that they are folded as an α -helix and bind in the cleft of Arp3 (Fig. 3b), which requires a couple of hydrophobic amino acids facing that cleft. In response to this comment, we added this clarification and also show the specific boundaries of the C and A regions in Fig. 1a.

Regarding Xie et al. PNAS 2021, we are not sure we can talk about a 'discrepancy.' First, as mentioned in our original manuscript, they analyzed C. elegans POD-1, which shares only 31% sequence identity with human Coro7 used in our study, and the differences are even greater toward the C-terminus, with POD-1 having an A region but no clear C region. Moreover, we do not describe any experiment—whether inhibition (Fig. 1) or debranching (Fig. 4)—that is directly comparable to theirs. Because the structure taught us that the Arp2/3 complex-binding region of Coro7 comprises both C and A regions, we analyzed

constructs that either had or lacked the CA region. Not knowing this, Xie et al. focused exclusively on the A region of C. elegans POD-1, which they deleted, and concluded that POD-1 FL and POD-1ΔA inhibited actin polymerization similarly in pyrene-actin assays and had similar debranching activity by TIRF. We do not have an equivalent ΔA construct to challenge their findings, although we suspect they are at least partially wrong, because binding of the A region to Arp3 should compete with NPF binding and thus contribute to inhibition. To conclude, we did different experiments on different proteins, and we believe it is better not to overstate the differences in the absence of firmer evidence to support disagreement.

3. The role of Coro7 in ER to Golgi trafficking that they claim as their main evidence for physiological function should be characterized more thoroughly. For example, does the Coro7 KO have overly stable patches of Arp2/3-branched actin on the ER and/or Golgi? Is this assay actually perturbed by inhibition of Arp2/3 function (e.g. treatment with CK666)? Since the F-actin binding is not very well defined (see point #1 above), we don't really know that this function depends on Coro7's ability to bind F-actin. For example, Coro7 binds to EPS15 and drives F-actin assembly on Golgi as well (see Yuan, Mol Cell, 2014), so their phenotype may have nothing to do with Arp2/3-debranching.

Indeed, linking the Coro7 KO phenotype specifically to debranching during ER-to-Golgi transport has been challenging. Yet, in response to this concern, we obtained new evidence supporting Coro7's involvement in this process. We co-imaged cortactin, a marker of branched actin networks, and GBF1, an ERGIC marker, in WT and Coro7 KO MCF10A cells (newly added Supplementary Fig. 7 and corresponding text highlighted yellow). We had previously shown that trafficking of GFP-GPI cargo through the ERGIC is substantially delayed in Coro7 KO RPE-1 cells (Fig. 5d-f). We now find that cortactin staining, which reflects branched actin, co-localizes with GBF1 in the perinuclear region of MCF10A cells and increases markedly upon Coro7 depletion, consistent with a defect in debranching. Together, these observations from two different cell types support the notion that ER-to-Golgi transport is impaired in the absence of Coro7.

We did not test Arp2/3 complex inhibition using CK-666 because both CK-666 and Coro7 expression reduce branch formation. In other words, a decrease in branching could not be specifically attributed to enhanced debranching by Coro7 constructs.

Reviewer 1

Reviewer 1 was fully satisfied with our response and changes to the manuscript.

Reviewer 2

The authors have addressed most of my concerns and the manuscript is almost ready for publication. One remaining thing: even with the new co-sedimentation assays, I still find the debranching model shown in Supplementary Video 8 highly speculative in the absence of any direct structural data. Statements about which propeller “likely binds first to the filament” are too strong, and should be replaced with “might bind first to the filament”. Structures will be required to obtain a more reliable model – I am happy to read in the rebuttal that the authors are pursuing this.

Regarding the suggestion to replace 'likely binds first to the filament' with 'might bind first to the filament,' we believe this comment refers to an earlier version of the manuscript, as the revised version does not specify which Coro7 domain binds first nor suggest priority among Coro7 interactions.

Finally, I may have missed it, but I also could not find titles and legends for all supplementary videos.

The reviewer is correct; as per Nature Communications instructions, supplementary video legends were provided in the Cover Letter. The journal further requests that these not be included in the Supplementary Materials.

Reviewer 3

The authors have addressed my criticism for the most part. I would still like to see them address the discrepancies between their findings and Xie et al (PNAS). This does not have to be particularly aggressive or confrontational statement, but just to acknowledge that they got different results and provide a possible explanation of why. This is basic scholarship 101.

The first paragraph of the Discussion addresses the discrepancy with Xie et al. (PNAS), namely our finding that the Arp2/3 complex-binding CA region of Coro7 is required for debranching, whereas those authors reported otherwise. For reference, we reproduce parts of this paragraph below, with the relevant text highlighted in yellow:

*“Both microfluidics-TIRF (Fig. 4) and cellular (Figs. 5 and Supplementary Fig. 7) experiments showed that all Coro7 domains must work synergistically to target branch junctions and promote their disassembly during ER-to-Golgi trafficking. Debranching was also previously observed with *C. elegans* POD-1 (28), indicating that this activity is conserved across Coro7-family members. Curiously, however, these authors found that the CA region was not required for debranching. It remains to be demonstrated whether this discrepancy reflects differences between POD-1 and human Coro7, which are only distantly related.”*